# RoboCT: The State and Current Challenges of Industrial Twin Robotic CT Systems

**DOI:** 10.3390/s25103076

**Published:** 2025-05-13

**Authors:** Gabriel Herl, Simon Wittl, Alexander Jung, Niklas Handke, Anton Weiss, Markus Eberhorn, Steven Oeckl, Simon Zabler

**Affiliations:** 1Technology Campus Plattling, Deggendorf Institute of Technology, Dieter-Görlitz-Platz 1, 94469 Deggendorf, Germany; simon.wittl@th-deg.de (S.W.); alexander.jung@th-deg.de (A.J.); anton.weiss@th-deg.de (A.W.); simon.zabler@th-deg.de (S.Z.); 2Fraunhofer EZRT, Flugplatzstraße 75, 90768 Fürth, Germany; markus.eberhorn@iis.fraunhofer.de (M.E.); steven.oeckl@iis.fraunhofer.de (S.O.)

**Keywords:** robotic CT, trajectory optimization, geometric calibration, CT reconstruction

## Abstract

Twin robotic X-ray computed tomography (CT) refers to CT systems in which two robotic arms are used to independently move the X-ray source and the X-ray detector around the object. This setup enables flexible CT scans by using robots to move the X-ray source and the X-ray detector around an object’s region of interest. This allows scans of large objects, image quality optimization and scan time reduction. Despite these advantages, robotic CT systems still face challenges that limit their widespread adoption. This paper discusses the state of twin robotic CT and its current main challenges. These challenges include the optimization of scanning trajectories, precise geometric calibration and advanced 3D reconstruction techniques.

## 1. Introduction

Modern-day industrial inspection validates quality metrics (sizes, surfaces, shapes) or searches for defects (holes, cracks, delaminations) in increasingly complex-shaped objects. In that respect, X-ray computed tomography (CT) belongs to the category of non-destructive testing and evaluation (NDTE) techniques [1]. CT was originally developed for medical scanning of human bodies [2]. Since 2001, CT has become an important tool for threat detection at airports. Meanwhile, CT is applied for inspecting the insides of partially or fully assembled products in the automotive and aerospace industries [3]. With the invention and development of micro-CT over recent decades [4], CT was further expanded to include testing batteries [5], as well as various electronic devices [6].

The fundamental property of CT technology is that volumetric material attenuation coefficients are computed from a finite set of ray-transmission measurements. These measurements are referred to as projections if pixelated flat-panel detectors are used. The rays intersecting in a single volumetric pixel (voxel) must originate from several directions, which in turn is realized by mechanical motion. The precision requirements for industrial CT and the stationary position of the patient in medical CT have significantly restricted the choice of mechanical actuators and, consequently, the range of available scanning trajectories. Conventional industrial CT systems typically rely on a turntable and a fixed source–detector geometry. These systems are well suited for small, symmetric objects but offer only limited flexibility. Large or asymmetric components often require view angles that cannot be achieved with rigid system geometries. In addition, full-object rotation may not be feasible for fragile, heavy or mounted parts. These limitations have motivated the development of robotic CT systems [7].

Robotic CT (RoboCT) refers to CT setups in which at least one component (source, detector or object) is moved by an industrial robot. A distinction can be made between mono robotic CT, where only one robot is used (e.g., to move a C-Arm with source and detector unit or the object) [8,9,10], and twin robotic CT, where both the source and the detector are mounted on separate robot arms. Figure 1 shows sketches of mono robotic and twin robotic CT systems to illustrate the key difference.

This paper focuses on twin robotic CT systems, also referred to as dual-arm robotic CT. In these systems, both the X-ray source and the detector are mounted on independent robot arms, allowing fully flexible and task-specific scan trajectories. These robots can be fixed to the ground [12,13], can be mounted on linear axes [14,15,16] or can even be mobile [17]. Furthermore, a turntable might be placed between both robots to additionally rotate the object [14]. Figure 2 shows the twin robotic CT system of the Deggendorf Institute of Technology, which consists of two robots placed on linear axes and a turntable.

This setup enables flexible CT scans, for example, to scan large objects [7,18], to reduce metal artifacts [19,20,21] or to shorten the scan time [22]. Currently, twin robotic CT systems are primarily used in the automotive [15] and aerospace [16,17] industries to scan large objects. Figure 3 shows examples of twin robotic CT systems scanning large objects.

Despite their flexibility, twin robotic CT systems have not yet been able to spread widely into industrial practice due to several challenges. The main challenges are CT trajectory optimization, geometry calibration, mechanical precision and 3D CT reconstruction.

CT trajectory optimization [21,23]: Depending on the application, various scanning trajectories yield CT volumes of different image quality. Furthermore, due to the size and complexity of the object, certain views may be inaccessible. For complex-shaped objects or ROIs with limited accessibility, the scan trajectory does not coincide with a circle or a spiral. Consequently, trajectory optimization algorithms must complement or, in some cases, replace decisions made by an expert operator.

Geometry calibration and mechanical precision [14,24]: For reconstructing sharp, accurate CT volume images from the above-mentioned set of projection views, the positions of the X-ray source and detector must be known with voxel/pixel precision. Depending on the required spatial resolution of an object or ROI, the positioning accuracy of industrial robot arms is generally insufficient to meet this requirement. Consequently, using single- or dual-arm robotic systems requires additional spatial calibration of the projection view geometry.

Three-dimensional CT reconstruction [8,25]: Dual-arm robotic CT systems enable imaging from non-circular and, in some cases, arbitrary trajectories. As a result, data processing and reconstruction algorithms must be capable of handling projections from such trajectories. Additionally, robotic CT is often used for large objects, requiring reconstruction methods that can process truncated data to reconstruct regions of interest (ROIs). While various CT reconstruction algorithms exist to handle these challenges, each comes with specific advantages and computational efficiency trade-offs.

This paper outlines the current state of the art and the challenges of twin robotic CT systems as well as their industrial applications. This review does not aim to classify different robotic CT system types or compare specific hardware implementations. Instead, it focuses on the fundamental methodological challenges, general principles and open research questions that arise in the development and application of twin robotic CT systems.

Section 2 provides a brief overview of the relevant definitions used in this paper, including the coordinate systems, scan poses, CT trajectory, aspects of robot integration and relevant CT artifacts. Section 3, Section 4 and Section 5 discuss the current main challenges of twin robotic CT systems, namely, CT trajectory optimization, geometry calibration and mechanical precision, as well as 3D CT reconstruction, along with state-of-the-art solutions for these challenges. Section 3 describes the challenge of finding the application-specific best views, i.e., the scan trajectory, within the boundary conditions imposed by the dual-arm robot and the object under investigation. Section 4 details the challenge of accurately estimating each projection view geometry, which is mandatory for an accurate CT volume reconstruction. Section 5 discusses various CT reconstruction algorithms applicable to data acquired by twin robotic CT systems. The section focuses on aspects that are particularly challenging for robotic CT applications, mainly arbitrary CT trajectories and ROI reconstruction of truncated data. The conclusion, given in Section 6, summarizes the overall capability and the residual error of state-of-the-art robotic CT systems when the aforementioned strategies are successfully employed. Overcoming these remaining limits is the subject of Section 7, which outlines future developments in the field of robotic CT.

## 2. Definitions

This section outlines the key concepts and terminology used throughout this document. We define the process of X-ray projection acquisition, describe how these projections are integrated with a robotic system to form a scan trajectory and explain the reconstruction process that converts the acquired projection data into a three-dimensional discretized volume representing the object’s internal attenuation coefficient function.

### 2.1. Coordinate Systems

A Euclidean coordinate system provides a framework for representing points in space using Cartesian coordinates in 3D. In robotics and computer vision, rigid transformations between coordinate frames consist of rotations and translations. These transformations are mathematically described by the Special Euclidean group SE(3), whose rotational component is given by the Special Orthogonal group SO(3) [26]. The group SO(3) consists of all $R^{3\times 3}$ rotation matrices R (with det⁡(R)=1) that represent pure 3D rigid body rotations. The group SE(3) represents full 3D rigid body transformations by combining a rotation and a translation. Specifically, an element of SE(3) is composed of a rotation matrix R∈SE(3) and a translation vector $\vec{t}\in R^{3}$ and is often represented by a homogeneous transformation matrix:(1)$$T=\left[\begin{array}{cc} R & \vec{t}\\ 0^{T} & 1 \end{array}\right]$$

This matrix describes the complete relationship between two coordinate frames. The coordinate system abbreviations used are as follows:W (world): Base coordinate system of the robotic system.O (object): Base coordinate system of the object or an ROI of the object that should be measured and reconstructed.D (detector): Coordinate system of the detector; the origin is in the center of the detector.S (source): Coordinate system of the X-ray source; the origin is in the focal spot of the X-ray source.

Therefore, for example, the transformation $T_{W}^{D}$ specifies the relationship of the detector coordinate system relative to the world coordinate system.

### 2.2. X-Ray Projection Acquisition

In industrial RoboCT, data acquisition involves capturing a series of X-ray projections from various viewpoints around an object. Each projection is associated with a specific scan pose, which encodes the positional and rotational relationship between the X-ray source and the detector. Formally, this scan pose, denoted by G, is represented as a tuple:$$G:=(T_{O}^{S},T_{O}^{D}),$$
with $T_{O}^{S},T_{O}^{D}\in SE\left(3\right)$, the transformations of X-ray source and detector in the coordinate system of the object. Given the object’s attenuation function$$f:R^{3}\to R,$$
and the scan pose G, a projection relative to the object coordinate system O is defined by the following operator:P(G,f)=I,
where I denotes the corresponding measured projection image. In practice, the measured projection image I is a discrete image sampled from the detector’s sensor array and is typically represented as$$I\in R^{w\times h},$$
where w,h∈N denote the number of pixels along the detector’s two spatial dimensions. A **CT trajectory** is defined as the set(2)$$T_{CT}=\left\{G_{1},G_{2},\ldots ,G_{s}\right\},\;s\in N$$
which represents multiple scan poses relative to the object coordinate system O, and where *s* denotes the number of views.

### 2.3. Robotic Integration

In order to execute the CT trajectory $T_{CT}$ with a robotic system, it is normally transformed into a **scan trajectory**$T_{Scan}$ in the world coordinate system *W*. We define a scan pose $\tilde{G}$ in the world coordinate system as$$\tilde{G}:=\left(T_{W}^{S},T_{W}^{D}\right),$$
such that the measured image of the projection operator relative to world coordinate system W (indicated by the tilde) remains the same:$$P\left(G,f\right)=P\left(\tilde{G},\tilde{f}\right)$$
Thus, the **scan trajectory** $T_{Scan}$ is given by$$T_{Scan}=\left\{{\tilde{G}}_{1},{\tilde{G}}_{2},\ldots ,{\tilde{G}}_{s}\right\}.$$
Each scan pose $G_{i}$, with i∈{1,…,s}, is mapped to a corresponding joint configuration vector through the robot’s inverse kinematics (IK) model:$$K_{inv}={\vec{q}}_{i},$$
where ${\vec{q}}_{i}\in R^{j}$ is a joint configuration vector, and j∈N denotes the number of joints of the robotic system. After obtaining joint configurations, each pose $G_{i}$ is evaluated to ensure that the system can physically attain it within its workspace and that the corresponding configuration is free of collisions. For collision checking, a model of the robot and the relevant collision objects, including the scan object, is employed. If a particular pose lies outside the robot’s reachable workspace, it is either excluded or modified to maintain feasible data acquisition. Since multiple IK solutions may exist for a given pose, only those configurations that are collision-free and satisfy the robot’s joint limits are considered valid.

Let $t_{1},t_{2},\ldots ,t_{s}$ be time instants at which these configurations should be achieved, with$$0\leq t_{1}<t_{2}<\cdots <t_{max}.$$
Then, the continuous robot trajectory is defined as$$Q\left(t\right):\left[0,t_{max}\right]\to R^{j},$$
with the following constraint:$$Q\left(t_{i}\right)={\vec{q}}_{i}=K_{inv}\left({\tilde{G}}_{i}\right),\;i=1,\ldots ,s$$
In other words, the discrete scan trajectory $T_{Scan}$ is mapped to a set of joint configurations $\left\{{\vec{q}}_{i}\right\}$ that are then connected with a smooth **robot path** to form the continuous **robot trajectory**
$Q(t_{i})$ over the time interval $\left[0,t_{max}\right]$, where $t_{max}$ equals the complete scan duration.

### 2.4. Reconstruction and Image Quality

Reconstruction in CT can be viewed as an inverse mapping that takes a set of acquired projections and converts them into a three-dimensional volume, thereby approximating the object’s attenuation function *f*. In this formulation, the reconstruction operator R is then defined as mapping the set of all projections to a discretized volume V:$$R(\left\{\left(G_{1},I_{1}\right),\left(G_{2},I_{2}\right),\;\ldots \;\left(G_{s},I_{s}\right)\right\})=V$$

Here, the continuous volumetric function *f*, which represents the attenuation coefficient at each point in the object, is approximated by a discrete set of voxels$$V\in R^{l\times m\times n},$$
with l,m,n∈N denoting the number of voxels along each spatial dimension.

It is important to note that, due to approximations in the inversion process, noise in the discrete measurements I or limitations in the number of projections, errors in the reconstructed volume may occur. These errors are commonly referred to as artifacts in CT imaging. Artifacts can manifest as distortions or spurious features in V that may obscure or alter the true internal structure of the scanned object.

The most common artifacts in robotic CT arise due to limitations in data acquisition, geometric inaccuracies and material properties. These artifacts, visualized in Figure 4, can degrade image quality and obscure critical details in the reconstructed volume. The choice of reconstruction algorithm plays a crucial role in the appearance and severity of these artifacts. For example, algorithms such as filtered back projection (FBP) or iterative reconstruction methods respond differently to issues like incomplete projections or misalignments. A deeper discussion on the reconstruction algorithms used in robotic CT, and their impact on artifact formation, is provided in Section 5.

**Limited-angle artifacts [27,28]:** These artifacts occur when the angular range of projections is restricted, resulting in incomplete data and reduced spatial resolution. This typically leads to blurred or distorted reconstructions. To generate these artifacts, projections were selectively limited to a subset of angles from the reference dataset.

**Sparse-view artifacts (undersampling) [27]** occur when the number of projection views is insufficient across the available angular range, resulting in streaking or aliasing effects due to undersampling. To generate these artifacts, fewer views were selected from a complete trajectory.

**Region-of-interest artifacts (ROI) [29]** arise when reconstructing a limited field of view, causing inaccuracies due to incomplete projection data outside the ROI. Reconstruction errors manifest at boundaries, leading to distortions or intensity variations. These artifacts were simulated by cropping the projections.

**Metal artifacts [30]** are generated by highly attenuating materials (e.g., metallic objects), leading to streak artifacts, beam hardening and distortions in reconstructed images. To simulate metal artifacts, a metal object was placed near the inspected object.

**Blurring artifacts [31]** occur due to random absolute positioning inaccuracies inherent in robotic systems. These errors introduce random deviations in scan poses, causing geometric uncertainty and reduced reconstruction accuracy. To simulate these artifacts, normally distributed positional noise was added to the scan poses. Typical examples of these errors, including their magnitude and characteristics, are provided in Section 4.

**Double-contour artifacts [31]** result from an incorrectly calibrated robot–tool geometry, specifically due to a constant offset in the source–detector alignment. Such errors cause systematic distortions, loss of spatial accuracy and double-contour artifacts. To simulate systematic geometric errors, a constant offset was applied to the scan poses.

Addressing these artifacts requires careful system calibration, optimized data acquisition strategies and advanced reconstruction algorithms to minimize their impact on image quality. These challenges are addressed in the following section.

## 3. Challenge: CT Trajectory Optimization

In RoboCT systems, monopolar X-ray sources with a maximum of 225 kV are used [7,14,25]. This limitation is due to the robots’ maximum load capacity and the weight constraints of the X-ray source and voltage cable. RoboCT systems are mainly used for the inspection of large industrial objects and the X-ray spectrum of the available sources is not always ideal. This can lead to a poor signal-to-noise ratio, especially with long and unevenly distributed attenuation lengths, resulting in attenuation artifacts in the CT images. The selection and parametrization of CT trajectories $T_{CT}$ can help mitigate these issues, but parametrization is often limited by the accessibility of scan poses G. As a result, only limited-angle CT trajectories are often feasible. This limits spatial resolution and can lead to limited-angle artifacts (see Figure 4). The FOV also depends on the selected CT trajectories and their parametrization. There are several methods for FOV extension, which either adapt a common CT trajectory or are new CT trajectories. The general challenge is therefore the identification and parametrization of a suitable CT trajectory that meets the requirements of the imaging goal while maximizing the flexibility of the RoboCT system. In the following, common types of CT trajectories used in industrial RoboCT systems will be reviewed. Their possible applications as well as their limitations will be shown.

### 3.1. Circular CT Trajectory

Circular CT trajectories—see Figure 5—were among the first to be realized for industrial applications with RoboCT systems [12]. The detector and source are positioned diametrically on concentric circles around the object’s field of view (FOV), with the ratio of their radii defining the magnification. The right-handed normal vector to these circles defines the axis of object rotation. This setup typically implies equiangular sampling. The scanning approach can involve either complete angular sampling (180° + source opening angle or multiples of 360°) or incomplete (limited-angle) sampling, depending on the imaging requirements. However, unlike conventional systems, the user can orient the plane and center of the circular CT trajectory relative to the object, allowing more flexibility without moving the object being scanned. In the context of RoboCT trajectories, the advantage of the circular CT trajectory is the low number of parameters of use and reliable results. Circular CT is most commonly used for scanning the region of interest, allowing for freely selectable orientations around the ROI. For an optimal circular scan, the goal is to cover an angular range of the ROI as large as possible. Strategies for the optimal placement of objects [32,33] developed for standard CT systems can be transferred to the RoboCT system and applied to select an optimized orientation of the circular CT trajectory. However, as previously mentioned, the circular CT trajectory is often limited by the dimensions of the scan object and the system, such as accessibility or attenuation. To address these challenges, strategies have been developed to optimize the parameters of circular trajectories. To account for the reachability constraints (see Section 2), Linde et al. [34] used an approach to adaptively select the focus distance of a circular scan. Butzhammer and Hausotte [35] showed that small adaptions of circular CT trajectories can be used to counter CBCT artifacts. By moving the source and detector on an undulated path around the circle (see Figure 6), the information is increased and CBCT artifacts are reduced—see [36]. One of the main advantages of the circular CT trajectory is that analytical reconstruction methods, such as FDK, can be used (see Section 5). This decreases the so-called ROI artifacts that can occur when reconstructing non-circular CT trajectories with truncated data. Therefore, if the kinematic restrictions allow for a circular trajectory around the ROI, it is often the preferred choice.

### 3.2. Computed Laminography

For scanning large, often flat objects at magnifications greater than one, circular trajectories are incomplete as they fail to capture the full long-side view of the object. An alternative approach is the use of a computed laminography trajectory, which also results in partial eclipsing of the long side. However, instead of limiting the angular range, CL tilts the axis of rotation relative to the optical axis, enabling improved coverage and reconstruction of the object’s features, as depicted in Figure 7. This CT trajectory helps mitigate beam hardening artifacts by maintaining a relatively constant transmission throughout the scan. Similar to circular CT, the CL trajectory used in RoboCT systems mirrors that of conventional laminography systems. However, the image quality in volumes obtained from CL trajectories is highly sensitive to geometrical parameters. Additionally, the selection of the imaging plane is important for CL trajectories, but this can be effectively managed with the flexibility of RoboCT systems, allowing for optimal plane selection to enhance image quality. One of the key benefits of RoboCT is its ability to enhance image quality, as highlighted by Rehak et al. [37], through the use of a spiral trajectory, as shown in Figure 8. In this trajectory, the detector moves in a spiral pattern within a plane, while the position of the source is defined by a line extending from the detector position through the scan center, maintaining a constant focus–detector distance (FDD). To prevent offset images in each projection, the relative orientation between the source and detector, as well as the FDD, is kept constant throughout the scan.

### 3.3. Field of View Extensions

To address the challenges of scanning large objects with RoboCT systems, particularly when larger FOVs are required, several advanced techniques have been developed. These methods aim to overcome the limitations of standard CT trajectories, which often restrict the ROI, especially in large-scale scans. The flexibility of RoboCT systems allows for the extension of FOV beyond traditional boundaries.

**Volume Stitching:** This method involves performing multiple scans where each scan’s projections are reconstructed individually, resulting in separate volumes. These volumes are then stitched together by blending the reconstructed slices to form a single, continuous volume. For successful merging, it is crucial that the acquisition geometries of all related scans match precisely.

**Projection Stitching [17,18]:** In this technique, as visualized in Figure 9, the field of view is expanded by virtually enlarging the detector area. This is accomplished by shifting the detector within the imaging plane while maintaining the beam geometry. As the detector shifts, it effectively covers a larger area, thereby increasing the field of view. This method is particularly useful when the object being measured exceeds the standard detector dimensions.

**Tomosynthesis/Combined Reconstruction:** This approach eliminates the need for stitching by incorporating the merging process directly into the reconstruction algorithm. It can be realized for either projection or volume stitching CT trajectories. By integrating data from multiple scans during reconstruction, it ensures a seamless and accurate representation of the object.

RoboCT systems are frequently tasked with scanning large objects. Advanced CT trajectories have been developed, such as the reverse helical CT trajectory in medicine. This approach, as illustrated in Figure 10, involves reversing the direction of the helix path upon reaching the specified acquisition angle α, allowing for the imaging of cylindrical FOVs with extended height *h*. For more complex, non-cylindrical FOV extensions, it becomes necessary to combine multiple CT trajectories. By integrating methods such as volume stitching, projection stitching and extended trajectories, RoboCT systems can effectively expand the field of view, enabling comprehensive imaging of large and irregularly shaped objects.

### 3.4. Arbitrary Views

The degrees of freedom (DoFs) provided by RoboCT systems allow the imaging process to be optimized across diverse industrial applications, accommodating objects of varying sizes, shapes and materials with high precision. Unlike conventional CT trajectories, which typically follow limited paths such as circular or linear scans, RoboCT systems can capture arbitrary views (as illustrated in Figure 11). This flexibility significantly enhances the imaging capability, enabling the development of optimized scanning trajectories tailored specifically to the geometry and properties of each inspected object.

Optimizing CT trajectories involves selecting an optimal subset $T_{CT}^{*}$ from an initial, larger trajectory set $T_{CT}$ (e.g., a spherical CT trajectory). The chosen subset should achieve either a higher reconstruction quality than standard CT trajectories or a shorter possible scan duration $t_{max}$. For the selection, a figure of merit M is used:$$T_{CT}^{*}=\arg\min_{T\subseteq T_{CT}}M\left(T\right)$$

The figure of merit M quantifies how well a trajectory $T_{CT}^{*}$ aligns with specific imaging goals. It might be linked to metrics such as the *detectability index* or other measures that directly assess the clarity and visibility of task-specific features [38]. Common metrics include Peak Signal-to-Noise Ratio (PSNR) [39], Structural Similarity Index (SSIM) [40], Mean Squared Error (MSE) [41], and Contrast-to-Noise Ratio (CNR) [42].

Due to the high degree of flexibility, it is impractical for a human operator to manually select arbitrary optimal scan poses. Hatamikia et al. [23] provide a comprehensive overview of CT trajectory optimization strategies. In the following discussion, we primarily focus on CT trajectory optimization within the context of industrial CT. We distinguish between two primary goals of CT trajectory optimization [21]:**Task-independent:** Task-independent trajectory optimization aims to improve the overall image quality without focusing on any specific task. This approach seeks to optimize the imaging process to generate the best possible images across the entire scanned area, ensuring that all features, regardless of their relevance to a specific task, are captured with high quality. These methods are beneficial when the imaging goals are broad, and there is no predefined task or feature that needs to be prioritized [19,21,22]. The optimization process in this case is more generalized, seeking to improve factors such as noise reduction, artifact minimization and spatial resolution across the entire image [43].**Task-dependent:** Task-dependent trajectory optimization is designed to improve the detectability of specific features or tasks within a CT scan [44]. The main focus is on optimizing the imaging process for a particular known task, such as identifying a specific region of interest or detecting certain features that correspond to crucial signals in the scan. This approach prioritizes the visibility and clarity of the task-relevant features, potentially at the expense of the overall image quality. For instance, some areas of the image might suffer from lower quality or increased artifacts, but the target task, such as detecting a specific anomaly, will be more easily distinguishable [45]. This method is particularly useful when the exact nature of the task is known beforehand, and the CT scan can be tailored to enhance the detection of those specific features.

Most CT trajectory optimization algorithms do not operate in real-time. This means that very detailed prior knowledge is required for the application of task-dependent optimization approaches. For example, detecting a crack or pore in a component requires prior knowledge of its exact structure, orientation and position to optimize the trajectory accordingly. In contrast, task-independent approaches aim to achieve a general improvement in image quality, which enhances the detection of unknown cracks or pores, without knowing their orientation or position. The significant advantage of task-dependent algorithms is evident in the reduction of the required number of projections [46]. For instance, if the side dimensions of an industrial component are to be measured, the algorithm can be specifically optimized for the surface, thereby reducing the number of required projections—see Figure 12. The quality of the CT optimizations is measured with various quality metrics.

In the following sections, we explore the possibilities of CT trajectory optimization in the context of industrial CT, with a primary focus on methods developed for RoboCT systems. However, we also consider approaches that, while not explicitly designed for RoboCT, could be readily adapted to these systems. We categorize the works based on whether they are task-dependent or task-independent and provide examples within these categories.

#### 3.4.1. Task-Independent CT Trajectory Optimization

**Data Completeness.** The basic idea behind data completeness is that if there are sufficient, high-quality data, reliable and correct reconstruction is possible. The figure of merit in this context quantifies the percentage of the necessary data that are available and meet the required image quality standards. Herl et al. [19] and Gang et al. [47] focus on optimizing CT trajectories by ensuring data completeness. Their method is based on the Tuy condition for data completeness. As stated by Tuy [48], for an object to be accurately reconstructed in three dimensions, every plane that intersects the object must also intersect the path of the source at least once. This ensures that sufficient information in Radon space is collected to theoretically reconstruct each voxel correctly. In their approach, the figure of merit is the percentage of necessary data that are available with good enough image quality. They incorporate local data quality metrics into the Tuy-based data completeness metric. Rays passing through highly attenuated areas are excluded, resulting in missing zones in the evaluation of the data completeness metric. The threshold for excluding these rays is chosen heuristically, based on empirical observations to balance the exclusion of unreliable data without discarding useful information. These missing zones are minimized by adjusting the CT trajectory to fill the information in these areas. Similarly, Gang et al. [47] apply the Tuy condition to design trajectories that reduce metal artifacts in CT scans. By ensuring data completeness, they aim to avoid missing spatial information, which can lead to artifacts. Their figure of merit quantifies the completeness of the data collected, taking into account the quality of the data. While Gang et al. [47]’s work is specifically aimed at reducing metal artifacts through geometric trajectory design for medical CT scans, Herl [21]’s approach generalizes the concept to ensure complete data acquisition specialized for industrial applications, thereby reducing artifacts and enhancing the overall quality of reconstructed images across a wide range of CT scenarios. Yuan et al. [49] build upon these foundations by introducing a machine learning-based approach to trajectory optimization. They employ Gated Recurrent Units [50], a type of recurrent neural network, to optimize scan trajectories in a non-greedy manner.

**Information Richness.** The fundamental idea of information richness is that the more information a projection contains, the better the reconstruction process can utilize it to improve image quality. Bussy et al. [51]’s work focuses on optimizing projection selection in sparse-view X-ray CT to improve image quality while reducing the number of required projections. They employ the Discrete Empirical Interpolation Method (DEIM) to identify the most informative projections based on a priori information about the object being scanned. The figure of merit is the information content of the projections, quantified using metrics like PSNR and SSIM. By selecting projections that contain the most information, they aim to enhance the overall image quality. Haque et al. [52] propose a spectral richness-based method to weight the information of a single projection, focusing on the frequency-domain properties of the projection data. The method uses a 2D Fast Fourier Transform on back-projected image data to evaluate the spectral power of each projection. Projections with higher spectral richness are considered to contain more information and are given higher priority. The figure of merit is the spectral richness, quantifying the information content in the frequency domain. In the works of Zeng [43] and Matz et al. [53], the figure of merit used for the selection of projections is based on a metric that correlates with the amount of edge information in a projection set. This metric assumes that the quality of 3D reconstruction is closely related to how well the projections capture the edges of the object. By quantifying the edge information, they aim to select projections that contain the most significant structural information. Entropy-based methods, as used by Dabravolski et al. [54] and Schielein et al. [55], utilize Shannon entropy [56] to measure the information content of projections and guide the adaptive selection of views. The assumption is that a higher information content, as indicated by entropy, enhances reconstruction quality, resulting in more accurate and detailed imaging.

#### 3.4.2. Task-Dependent CT Trajectory Optimization

**Testing Task-Specific Simulations.** The basic idea is that if a scan yields high image quality in a realistic simulation, or if a given task can be successfully accomplished within the simulation, then it is likely that the same task can be performed in a scan—assuming the simulation is sufficiently accurate. The figure of merit is determined by evaluating the image quality achieved in the simulation or assessing the effectiveness with which the task is completed within the simulated environment. Brierley et al. [57] developed an algorithm that selects projections by incorporating prior knowledge about the component being inspected, including its geometry, material properties, expected defect locations, and critical regions. The algorithm uses computer simulations of the inspection process to predict the output for a given configuration of simulated projections. The figure of merit is based on the simulation’s ability to accurately perform the task (e.g., defect detection). By iteratively optimizing the set of projections to maximize defect detectability in the simulation, they aim to ensure that the task can be successfully completed in the real scan. Schmitt et al. [32] optimized the placement and orientation of the workpiece within the CT system to ensure the best possible measurement accuracy. The algorithm uses ray-tracing simulations to evaluate different workpiece orientations and calculates the corresponding X-ray projections. The figure of merit includes metrics such as contrast-to-noise ratio (CNR) and measurement uncertainty in the simulation. By choosing the orientation that yields the highest image quality in the simulation, they aim to achieve optimal results in the real scan.

**Model Observers.** The fundamental principle behind using the detectability index is that if the data are sufficient for a model observer to perform the task, then the CT scan is sufficient for the task. The figure of merit involves estimating whether a model observer could perform the task based on the Modulation Transfer Function (MTF), the Noise Power Spectrum (NPS) and the specificities of the task itself. In the paper by Gang et al. [44], the figure of merit used for optimizing the selection of projections is the detectability index. This index evaluates how well a model observer can detect and differentiate relevant features in the presence of noise and artifacts. The detectability index is expressed as a function of the imaging task, the MTF and the NPS. By estimating whether a model observer could perform the task based on these parameters, they select projections that maximize task performance. Bauer et al. [22] applied a similar approach by focusing on optimizing CT trajectories to enhance the detectability of specific features. They utilized the detectability index as the figure of merit, ensuring that the selected projections contribute to achieving high-quality image reconstruction for the task at hand. Zaech et al. [45] used a deep convolutional neural network (ConvNet) to predict the detectability index for each potential next view based on current and previous 2D X-ray projections. By dynamically adapting the trajectory to maximize the detectability index, they aimed to enhance the task-specific image quality in real-time. Schneider et al. [58] extended this work by optimizing CT trajectories in a twin robotic CT system for the application of pores in casting parts. In [20], the metric used to evaluate and optimize CT scan trajectories is a combination of two local metrics: the detectability index and a Tuy-based measure of data completeness. By combining these metrics, they aimed to optimize both data quality (detectability) and completeness simultaneously. The figure of merit quantifies the amount of necessary data that are available for the task, ensuring that the selected trajectory maximizes both the detectability of the features of interest and the completeness of the data for accurate 3D reconstruction.

### 3.5. Conclusion: CT Trajectory Optimization

The optimization of CT trajectories offers a wide range of methods, particularly concerning various quality measures, to achieve diverse objectives. These methods can provide significant advantages in terms of image quality [19,47], task detectability Bauer et al. [22], Zaech et al. [45] and reductions in scan time [46]. Despite these advancements, several challenges need to be addressed. A primary concern is the need for prior knowledge to implement most methods effectively. Except for the work by Zaech et al. [45], there are currently no approaches for live optimization during the scanning process. Additionally, the computational effort is often high, especially when volume data must be computed, which limits applications in real-time scenarios. Furthermore, there is a lack of systematic comparisons between different methods, making the selection of the appropriate technique difficult. Choosing the right method, particularly the quality measure, often requires expert knowledge and can involve trial and error. Since the quality measures are mostly based on heuristics, it is not always apparent when and to what extent they function reliably. There is potential here to further optimize the heuristics. Despite the recent advancements by Linde et al. [59], most available methods are limited to ‘standard basic sets’ like spherical CT trajectories and do not yet support fully arbitrary scan positions. In the work of Bauer et al. [46], the scan positions are not limited to a set but limited to a plane. Additionally, precise preparation is required, including accurate object positioning and adherence to safety margins. In summary, although CT trajectory optimization offers significant benefits in certain areas, it still faces challenges that require further research and development. Future work should focus on developing methods for live optimization, reducing computational effort, conducting systematic comparisons and extending applications to arbitrary scan positions.

## 4. Challenge: Geometric Calibration


Conventional industrial CT systems are limited in the variability of the position and orientation of the X-ray tools. Robot CT systems use industrial robots as flexible manipulators to overcome this limitation. However, the absolute accuracy of industrial robots is often several orders of magnitude lower than that of the manipulators in conventional CT systems. Using the notation defined in Section 2: When commanding the robots to move to the scan position $\tilde{G}=\left(T_{W}^{S},T_{W}^{D}\right)$, they instead reach a slightly different, unknown position ${\tilde{G}}_{error}=\left(T_{W}^{S_{error}},T_{W}^{D_{error}}\right)$ due to unmodeled deviations. These deviations depend on the specific robot model, as described in the conclusion of the robot calibration section (Section 4.1.5), and are typically in the order of a few millimeters. This positional inaccuracy is up to two magnitudes higher than is required for an accurate reconstruction. This initially limits the use of robots in CT applications, as CT reconstruction requires precise information about each scan pose. This restriction can be addressed through several calibration procedures. In the context of X-ray CT and metrology, calibration refers to the process of determining and rectifying the deviation of a measuring instrument from a standard [60]. In this case, this refers to the difference between $\tilde{G}$ and ${\tilde{G}}_{error}$. In the CT reconstruction process, incorrect information about the poses of X-ray source and detector can lead to various image artifacts in the 3D volume such as double contours and blurring. A detailed description of possible geometric errors and their effects on reconstruction can be found in [31].

Figure 13 compares CT reconstructions acquired with the uncalibrated robot CT system of the Deggendorf Institute of Technology—where geometric errors lead to positional deviations of roughly 1.5 mm—with those obtained after calibration via the Direct Linear Transformation (DLT) method, which reduces the residual error in the image plane to below 0.2 mm [14]. Various calibration approaches, including classical robotic calibration, imaging methods and calibration objects (further detailed in this work), were evaluated in [61] for their impact on reconstruction quality using metrics from both spatial and frequency domains.

This section is divided into two main approaches for the geometric calibration of robot CT systems. Section 4.1, Robot Calibration, details methods for calibrating robots without considering the X-ray data. Section 4.2, Imaged-Based Calibration, outlines methods for using the X-ray data itself to determine the poses of the X-ray tools.

### 4.1. Robot Calibration

In robotics, the term *calibration* encompasses the calibration of the robot itself, the determination of a reference point on the tool (tool center point, TCP) and the robot pose relative to a global coordinate system (environment calibration).

The purpose of environment calibration is to determine the relative position and orientation of the robot and objects involved in the process within a common reference coordinate system, e.g., world. Tool calibration establishes the relationship between the tool’s coordinate system and the flange coordinate system of the robot’s last axis. In the context of a robotic tomography system, it is necessary to determine the correct position and orientation of the focal spot of the X-ray source and the detector’s imaging plane.

Robot calibration is performed to improve pose accuracy (static calibration) or to enhance path accuracy (dynamic calibration) [62]. Depending on the application, the process-relevant target metric must be taken into account. In computed tomography, data are usually acquired with the X-ray tools in a static position while the sample is being imaged. Therefore, only the accuracy of the positioning contributes to the measurement result. Static robot calibration is therefore of primary interest.

Typically, a robot’s pose is approximated using either a kinematic or a non-kinematic model of varying complexity. While the most basic formulation describes the robot by its nominal geometry, more sophisticated models also take linear effects (e.g., manufacturing, assembly errors [63,64]) or non-linear effects (e.g., gear-backlash, thermal expansion, elastic deformation [65,66,67]) into account. Independent of the actual choice of model parameters, most model and calibration approaches follow the same principle. The key parameters influencing pose accuracy must be identified and appropriately described using mathematical models. Finally, a suitable data acquisition approach and, if necessary, measuring device must be chosen.

Based on these steps, the subsequent subsections highlight some of the more prominent examples of research on calibration methods, with a focus on their application to serial kinematic chains. The body of research in this field is extensive, as reviewed in the literature of about 40 years [64,65,68,69,70,71], particularly regarding vertical articulated robots, which are emphasized due to their significant industrial relevance.

#### 4.1.1. Factors Affecting Pose Accuracy—Error Sources

The absolute accuracy of industrial robots is typically significantly worse than their repeatability [63]. The maximum deviation of the pose repeatability can be assumed as the lowest threshold of the absolute accuracy, only limited by stochastic errors. In contrast, absolute accuracy is affected by a range of mechanical errors and furthermore physical influences, which can be classified into manufacturing or operational influences, as well as systematic or stochastic effects.

Influences that impact pose accuracy can be broadly categorized into elastic deformations, temperature effects and manufacturing-induced tolerances. Elastic deformations within the robot’s structure, such as those occurring in the kinematic chain or within harmonic drives, can introduce considerable deviations from the expected pose. Temperature variations can alter the geometry of components, affecting accuracy over time [65,66]. Manufacturing-related factors, including tolerances, assembly errors and material properties, contribute to systematic errors between the robot’s actual performance and its nominal model.

**Kinematic** influences are predominantly related to manufacturing deviations and are typically characterized as linear offsets. These include geometric errors in robot components, such as zero-point errors and assembly inaccuracies, which generally manifest themselves as systematic errors. Although these kinematic errors are systematic and can be corrected through offset determination, their influence on pose accuracy is significant [63].

**Non-Kinematic** influences, such as gear backlash and temperature effects, introduce non-linear deviations that are more complex to model. These factors must be carefully considered depending on the robot’s configuration. For example, the elastic deformation of harmonic drives can be interpreted as contributing to gear backlash, leading to additional pose inaccuracies. Non-geometric influences, such as torsional compliance, temperature-induced link length variations and gravity, also play a significant role in reducing pose accuracy. These non-linear effects require advanced models to accurately model and mitigate their impact on the robot’s overall performance [65,66,67,72].

#### 4.1.2. Calibration Data Acquisition

External systems used to acquire calibration data for a robot typically include laser trackers, optical coordinate measurement systems, articulated arm coordinate measuring machines (AACMMs) and tactile measurement systems.

**Laser trackers** provide high-precision (e.g., API Radian 15 μm + 5 μm/m^−1^) volumetric accuracy [73] and real-time measurements of the robot’s end effector position within a defined space. Due to the underlying measurement technology, a line of sight is required to a passive or active retroreflector. This can impose a limitation on the available measurement volume.

**Optical coordinate measurement systems** utilize cameras to track the spatial position and orientation of optical markers on the robot’s components with high accuracy (Carl Zeiss aG, GOM ATOS5, no exact precision specified).

**Articulated arm coordinate measuring machines**, equipped with precise encoders and mechanically coupled to the robot, offer direct measurement of the robot’s end effector pose, allowing for the capture of detailed kinematic data that significantly enhance overall calibration accuracy (FARO FaroArm, up to 24 μm [74]). Through coupling a secondary mechanical system to the robots, the robots’ flexibility can be restricted. The addition of more moving mechanical components also increases the complexity of collision detection and robot path planning.

**Tactile measurement systems**, using a tactile probe, enable precise contact-based measurements, though they are spatially limited by the reach of the probe and the necessity of physical contact with the robot or reference objects (Renishaw MP250 [75], 3D measurement deviation down to 1 μm). This measurement method requires a calibration artifact, and depending on the calibration procedure, also its position. This limits the usability of touch probes for CT applications, but is of use in machining tasks.

**Motion capture systems** use multiple monocular cameras to triangulate markers within the measurement volume. The markers are either passive reflective markers or active infrared light-emitting diodes. The system’s performance highly depends on the number of cameras used, which affects their ability to mitigate occlusion and false detection by reflective surfaces. The smallest detectable motion can be down to 70 μm (Qualisys Miqus M5 [76]). Such systems frequently use flashed infrared illumination, which can limit the use of other optical measurement technology.

**Drawstrings** use a string on one side connected to the robots TCP, and on the other side connected to a known reference point. The length of the drawstring is measured with high precision (e.g., 4 μm demonstrated by Li et al. [77]). Additional strings can be used to allow for triangulation as implemented by Dynalog Inc. [78].

#### 4.1.3. Calibration Procedures

Calibration approaches for robotic systems focus on minimizing the discrepancy between the measured and reported end effector positions or full poses to enhance accuracy. The most commonly used and straightforward optimization methods include Least Squares and Levenberg–Marquardt algorithms [79,80]. These methods adjust the robot’s model parameters to align the model’s predictions more closely with actual measurements. Optimization algorithms iteratively refine these parameters to achieve the best possible match between the model and the observed data, while least squares minimization specifically targets reducing the sum of squared differences between predicted and measured values.

The resulting parameters from these methods are designed to minimize the overall systematic error without relying on other variables. They are optimized to reduce the total error, independent of spatial factors or non-kinematic influences, thereby ensuring a more generalized and robust calibration. The source of the actual system’s pose or position is arbitrary, with the choice of reference measurement system often depending on the laboratory’s available equipment. Laser trackers are typically the preferred choice due to their high precision, but are a rather expensive. Furthermore, other reference systems may be used on the basis of specific experimental setups.

When addressing **thermal effects** during calibration, it is crucial to actively excite the robot through movement to account for temperature variations that may affect its structural components, as performed by Le Reun et al. [66] and Sigron et al. [65]. This procedure ensures that the calibration process can identify and correct thermal-induced errors. In this context, the links with the most significant influence on the end effector position must be identified and corrected, as they are more susceptible to thermal expansion or contraction.

When trying to identify **gear backlash**, calibration must focus on identifying the axes with the most substantial errors due to this effect. This requires careful analysis of joint behaviors, particularly the direction in which joints approach their positions, as this factor is critical in determining the extent of backlash. Furthermore, the dependence of this error correction on the rotation direction of the joints makes it more complex to integrate within a robot abstraction model [66,81,82].

**Joint compliance**, which depends on the load applied to the joint, is a significant source of error in many robotic systems. It can be measured by applying external forces while clamping other axes to isolate the deflection of the joint under test, or by disassembling the robot and measuring each joint separately. These methods are particularly relevant in applications involving dynamic loading, such as machining. However, in scenarios where motion is slow and the load remains constant—such as in this study—these techniques are likely unnecessary [83].

**Data-driven** approaches mostly ignore specific modeling of physical effects. The general approach is to define geometric offset parameters, which are to be determined by machine learning [84,85,86].

#### 4.1.4. External Prismatic or Revolute Joints

To further increase the manipulability of an industrial robot, it can be mounted on a linear rail (prismatic joint), as shown in Figure 14. Its influence on positional accuracy is not negligible, as shown by Borrmann and Wollnack [87]. Especially the rotational deviation depending on position can introduce significant non-linear deviations (greater than 0.1 mm).

Additionally, as included in conventional CT machines, the system can be equipped with a rotational stage. In our case (see Figure 14), a rotary indexing ring is used, of which the positional errors are negligible in comparison with other axes in the system (axial and radial runout of 30 μm [88]).

#### 4.1.5. Conclusion to Robot Calibration

Depending on the literature and the process for which the calibration is being investigated, the most suitable method may differ. Le Reun et al. [66] investigated the pose accuracy of ABB IRB 4600 robots in the context of a twin robotic CT system. The approach chosen is a model considering geometric, thermal and gear backlash effects, resulting in a pose accuracy improvement from 1.53 mm to 0.07 mm. Sigron et al. [65] considered the same effects in an arbitrary setting, while performing the experiments on a KUKA KR 16 and KR 30. The average error after full calibration was reduced from 0.36 mm to 0.12 mm in the case of the KR 30 and from 3.39 mm to 0.13 mm in the case of the KR 16. Interestingly, both robots had a mean error of about 0.2 mm after the kinematic calibration only. As mentioned in Section 4.1.3, especially the thermal calibration requires a relatively long excitation period, so the effects of motor heating can be measured on the robot links.

Nubiola and Bonev [63] considered geometric and elastic effects of an ABB IRB 1600 robot. In this case, the average error was improved from 0.968 mm to 0.364 mm. In this case, the dataset for calibration and validation of the robots’ performance was extensive, with 1000 poses overall. In the case of Klimchik et al. [67], a geometric and elastic approach was chosen for a KUKA KR 270, since the robot was subjected to dynamic loads due to machining tasks. In their case, the maximum positional errors were corrected from 3.5 mm to 0.4 mm, highly depending on the position within the working area.

Lee et al. [89] performed a geometric calibration on an unspecified robot model, with an improvement from about 0.8 mm to 0.2 mm. The results show a significant improvement, but were validated on a small set of robot poses.

Despite the lack of direct physical modeling, Cai et al. [84] reduced the maximum error of a KUKA KR 120 2700 robot from 1.6866 mm to 0.3565 mm by using an extreme learning machine approach trained on 1000 random robot poses.

This excerpt from the overall research of industrial robot calibration serves as an illustration of the large differences between various influencing factors on the absolute accuracy, as well as their characterization. On the way to robot-based dimensional accuracy, there are various challenges to overcome, ranging from the choice of a suitable robot candidate to the selection of appropriate calibration procedures to achieve the required precision.

### 4.2. Image-Based Calibration

In contrast to robot calibration, image-based calibration uses the information from the X-ray projections to determine the geometric information. In theory, image-based calibration is therefore completely independent of the manipulator system.

In this work, we categorize image-based calibration methods into two distinct approaches: *Offline Calibration* and *Online Calibration* methods. Offline calibration requires a secondary scan specifically for geometric calibration, whereas online calibration relies solely on the data from the scan of the relevant region of interest of the specimen.

#### 4.2.1. Offline Calibration

The repeatability of industrial robots often exceeds their absolute accuracy by several orders of magnitude [63]. Offline calibration methods take advantage of this by performing a second scan of a calibration phantom, enabling precise geometric calibration. The general workflow is as follows: First, a CT scan of the relevant region of interest of the specimen is performed with the trajectory of choice. Second, a CT scan of a calibration phantom at the same spatial location is performed with the same trajectory.

Based on the data from the additional scan, image processing methods compute the geometric information for each individual projection. Commonly used techniques include the DLT [90] and ellipse fitting methods [91]. DLT calculates the geometric information relative to the coordinate system of the calibration body and can be applied to arbitrary trajectories. In contrast, ellipse fitting methods are limited to circular trajectories. The results from these calculations, the geometric information of the calibration scan, are then used in the reconstruction of the CT scan of the relevant specimen.

Phantoms for this offline calibration workflow mostly consist of easily penetrable material and several highly absorbing spheres, in the following referred to as markers. Spheres are used as markers due to their invariance under rotations and the availability of established methods for precisely determining the center of projected spheres.

Existing methods can be categorized into methods using phantoms with known marker geometry, in the following referred to as calibration bodies, and phantoms with unknown marker geometry. The geometry of calibration bodies is determined in advance using tactile measurements, precise manufacturing or high-resolution CT scans with a calibrated system.

**Offline calibration using a calibration body:** There are several types of calibration bodies, each differing in the positioning of the markers. The differences result from the type of CT system and the requirements resulting from the trajectories to be calibrated. A plastic cylinder with 24 steel ball bearings precisely located in two circular patterns was used in [92], whereas [14,93,94] used plastic cylinders with spiral assignment of the markers. The calibration body used in [14] is shown as an example in Figure 15a. A similar spiral arrangement, but with an additional larger-diameter marker at the top center, is used in [95]. Ref. [96] used two parallel plastic planes with 44 markers. Ref. [97] used fourteen markers arranged in a detailed configuration: four along each of three perpendicular axes and two along a diagonal axis.

**Figure 15 sensors-25-03076-f015:**
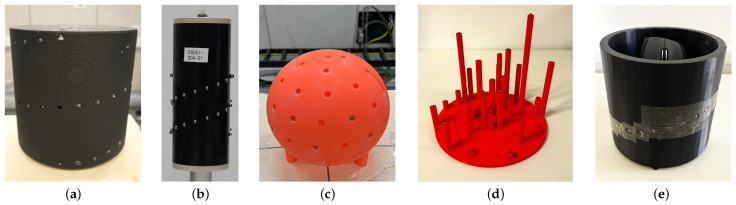
Calibration bodies made of plastic with spherical metal markers. Bodies (**a**,**b**,**e**) utilize a spiral arrangement of the markers, which is particularly well suited for CT trajectories that deviate only a few degrees from an ideal circular trajectory. Figure (**c**) shows a spherical calibration body with a uniform spherical distribution over the surface, which allows it to be used for arbitrary trajectories. Figure (**d**) shows a calibration body that contains multiple groups of four collinear markers. These markers are automatically identified based on their projective cross-ratio, which enables fully automated calibration for arbitrary trajectories. Bodies (**a**,**c**,**e**) are used for the CT system shown in Figure 2, (**b**) for the Hexapod CT presented in [35] and (**d**) for the twin robotic CT system from BMW. (**a**) Calibration body used in [14]. (**b**) Calibration body used in [35,98]. (**c**) Spherical calibration body used for arbitrary trajectories in [99]. (**d**) Calibration body as described in [100]. (**e**) Hollow calibration body carrying a mouse as described in [101].

Conventional CT systems upgraded with a hexapod (**Hexapod CT**) represent a special case of a robotic CT system. Butzhammer et al. [35] used the intrinsic parameters of the underlying CT system and adapted the method from [96]. They reduced the calibration task to finding the translation and rotation from the coordinate system of the turntable to the coordinate system of the hexapod. The calibration body used there is shown in Figure 15b. Weiss et al. [61] presented a method in which only 40 of the previously acquired projections are repeated, while the missing projection geometries are interpolated based on the nominal trajectory on a 12-DoF system.

All of the mentioned offline calibration methods involving a calibration body require the correct assignment of projected markers to their corresponding positions on the calibration body. This process often demands manual intervention, which [99] automated by employing AI for marker assignment. The spherical calibration body used in the foregoing study for arbitrary trajectories is shown in Figure 15c. Another approach to automating the marker assignment was first introduced by [100] and later implemented in [102] through the use of a specialized calibration body. The calibration body consists of multiple groups of four markers arranged linearly. The detection and matching is based on the cross-ratio of the markers along the lines, which allows automation of the assignment. An example of such a calibration body is shown in Figure 15d.

**Offline Calibration Using a Phantom Without Known Marker Positions:** This section describes methods that use phantoms with unknown marker geometry. All of these approaches use the entire stack of projections as initial information. The absence of known marker geometry initially complicates the calibration process.

Consequently, the majority of these methods are designed for systems with fewer degrees of freedom, relying on the assumption of perfect rotational movement. As such, many of these approaches are suited to conventional industrial CT systems with turntables. Nonetheless, these methods are relevant in this work due to their potential adaptability to robotic CT systems.

Hardner et al. [103] and Li et al. [104] proposed methods that detect and track markers of unknown position across the projections and utilize a non-linear least squares optimizer to adjust geometric information to fit a model. Smekal et al. [105] suggested a method based on a Fourier analysis that also works with the course of the markers on the projections. Another analytically motivated auto-calibration approach, which, however, optimizes the projection matrices of each projection and the marker arrangement, was proposed in [106]. Assuming perfect rotation of the rotation stage, ref. [107] derived a formula to establish the relationship between the ellipses fitted to the projection image curves of the markers and the geometric information.

In the case of Hexapod CT, Butzhammer et al. [108] presented a method based on imaging a single metal sphere that is moved to various positions by the hexapod to determine the coordinate transformation from the coordinate system of the turntable to that of the hexapod. Blumensath et al. [109] presented a method using one or several markers that are moved by a known amount along one degree of freedom. A multi-step optimization approach uses the assumed positions to find the intrinsic parameters of their Hexapod CT as well as the alignment of the linear and rotational manipulator axes.

Some methods for offline calibration without known marker positions also apply to arbitrary trajectories: Li et al. [110] introduced a method that first determines the precise intrinsic geometric relationships of the markers within the phantom, followed by an iterative calibration process using optimization constraints based on a back-projection model. Later, Li et al. [111] proposed a more advanced model in which the parameters to be optimized can be significantly reduced, thereby reducing the computing time while maintaining stability against noise.

#### 4.2.2. Online Calibration

Online calibration methods require only a single CT scan. This is more efficient, but means that the geometric information has to be extracted directly from the projections of the CT scan with the specimen in place. In this section, we distinguish methods that use prior knowledge of the object and methods that function without any prior information.

**Online calibration using prior knowledge**: A method to avoid an extra calibration scan while still utilizing a calibration body was proposed by [10,101]. In their approach, the specimen is placed inside a hollow calibration body so that both the specimen and the calibration body can be scanned simultaneously. An example of this setup is shown in Figure 15e. This technique is limited to specimens that can fit within a calibration body. Butzhammer et al. [112] evaluated this type of trajectory calibration in comparison to offline calibration using a calibration body to determine the influence of the pose repeatability.

Other online calibration methods extract geometric information by optimizing image quality metrics or aligning projection images with simulated forward projections derived from a known specimen model. This model can be based on a CAD design or a prior scan, a process known as 3D-2D registration.

Ouadah et al. [113] and Chung et al. [114] used forward projections of CT reconstructions previously acquired on conventional CT systems to compare them with projections acquired by a C-arm. Ouadah et al. were able to use their method for arbitrary trajectories, whereas Chung et al. only focused on tomosynthesis. Both used a linear optimization algorithm that minimizes an objective function that compares the forward projections and actual projections while taking all geometric parameters into account. Their methods achieved comparable results to an offline calibration method used for C-arm systems already earlier described [92]. Tonnes et al. [115] separated the optimization of rotations and translations, achieving similar results to the 3D-2D registration methods mentioned before, but faster and with higher tolerance to errors in the initially guessed parameters. Ji et al. [116] presented a different 3D-2D registration approach to C-arm CT calibration by utilizing a non-linear registration model to address complex system distortions. The non-linear model accounts for more complex distortions in the C-arm CT system that cannot be captured by linear methods, e.g., the bending of the C-arm. Bussy et al. [117] proposed a 3D-2D registration between X-ray projections and a CAD model.

Kyriakou et al. [118] proposed an iterative calibration method for approximately circular trajectories. In each step, a simplex algorithm minimizes the entropy of the volume, thereby optimizing the image quality by refining the geometric information.

Furthermore, there are methods utilizing artificial intelligence (AI). For calibration of the rotation axis, Yang et al. [119] proposed a classical convolutional neural network approach, while Presenti et al. [120] used a feature extractor. Disadvantages of both methods are high computational costs and the need to train the network for every new specimen.

**Online calibration using no prior information:** There are two types of approaches for online calibration without using prior information about the specimen. In the first approach, markers with unknown positions are attached to the object. The 3D positions of these markers are estimated and then used for calibration, similar to offline calibration methods. In the second approach, no markers are used and calibration is performed solely based on the projection data of the specimen.

Marker-based online calibration methods use markers with unknown positions placed on the specimen. To clarify the principle of determining the positions of these markers, let us first consider one single marker *k*, as illustrated in Figure 16. For every projection, there exists a ray connecting the X-ray source, the marker and the marker’s image on the detector. With accurate geometric information, all rays from all projections would intersect at the actual marker location $p_{k}$, as can be seen in Figure 16a. However, with inaccurate geometry, these rays do not necessarily intersect. In this case, the position of the marker has to be estimated, as can be seen in Figure 16b. The specific methods for identifying and optimizing these positions vary.

Bossema et al. [121] enabled the use of in-house 2D X-ray equipment in museums to perform 3D CT scanning. Their method begins with educated guesses of the marker locations, followed by a least squares optimization for calibration. Ma et al. [122] used the imprecise geometric information from the CT system as a starting point and computed the mean of all virtual intersections between each pair of rays, as described in Figure 16. By virtual intersection, they mean the point with the shortest distance to a pair of skew 3D lines. A non-linear optimizer is then applied to further refine both the estimated marker locations and the geometric parameters.

**Figure 16 sensors-25-03076-f016:**
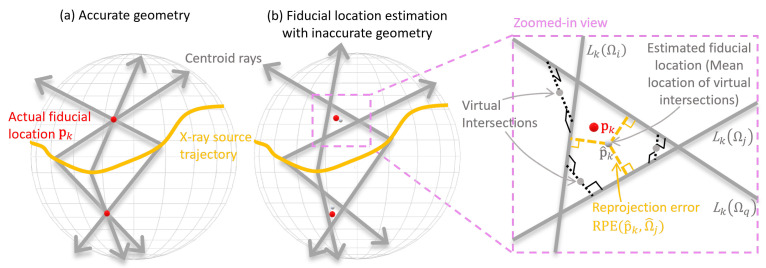
Diagrams of backprojections of fiducial markers under (**a**) accurate geometry and (**b**) inaccurate geometry. Reprinted, with permission, from [122].

These marker-based online calibration methods have the disadvantage that one needs to adjust markers to the specimen, which slows down the scanning process and can lead to marker-related metal artifacts in the reconstruction.

Marker-free online calibration methods that do not rely on prior information are based on the assumption that a correctly calibrated CT system should produce consistent scan data. This requires the absence of significant artifacts and that the entire specimen is fully captured within the field of view of all projections. These constraints limit the applicability of these methods in general robotic CT calibration scenarios, where scans of large objects, heterogeneous materials and partial object scans are often necessary.

Meng et al. [123] exploited the sum of all projections to refine geometric calibration. Debbeler et al. [124] introduced a projection redundancy criterion to quantify inconsistencies in projection data, employing a global optimization of multiple geometric parameters, such as detector shifts, to enhance CT alignment. Building upon Grangeat’s fundamental relation for cone-beam CT [125], Maass et al. [126] extended Debbeler’s method by adapting geometric parameters locally on a per-projection basis. Cant et al. [127] further refined this approach to improve robustness in limited-angle tomography.

A significant advancement in consistency-based calibration is the epipolar consistency condition, initially proposed by Maass et al. [126] and later formalized by Aichert et al. [128]. The concept is rooted in epipolar geometry, which describes the geometric relationships between multiple images of the same object taken from various viewpoints. The core principle is that a line integral along a given ray path should yield identical values across multiple projections of the same object. Aichert et al. [128] demonstrated how the epipolar consistency condition can be applied for calibration purposes and later extended its use to 3D patient tracking [129]. Frysch et al. [130] adapted this methodology to C-arm short scans, specifically for cone-beam geometries with semicircular trajectories.

As a data fidelity-based alternative, Li et al. [131] proposed an iterative brute-force method that maximizes the consistency between projection data and the reprojected counterparts of a reconstructed volume. In [132], Li et al. further improved their geometry correction approach by incorporating a normalized cross-correlation metric, enhancing the accuracy of robotic CT region-of-interest imaging. This advancement made it possible to achieve more precise geometry correction in cases where full object coverage was not feasible. A common limitation of current methods is that they provide only point estimates of the corrected projection geometry, without offering any measure of uncertainty. To address this, Pedersen et al. [133] proposed a Bayesian-based method that jointly estimates the projection geometry and reconstruction while also providing uncertainty estimates. Numerical experiments demonstrated that their method performs similarly to current geometry correction techniques with high-quality, low-noise data, but outperforms them under more challenging conditions, such as with fewer projections or highly noisy data. Additionally, it can be generalized to other CT setups, including parallel-beam and 3D CT.

Other approaches that calibrate and reconstruct at the same time use comprehensive AI. Rückert et al. [134] proposed an adaptive, hierarchical neural rendering pipeline for multi-view inverse rendering based on deep learning. Their Neural Adaptive Tomography framework (NeAT) is a further development of the Neural Radiance Fields (NeRF) framework [135] with the focus on CT reconstructions. The projection geometry of the scan is a parameter in their framework that can be adjusted, allowing geometric errors to be determined through backpropagation. In [136], a differentiable forward projector was used for calibration. The method’s performance was evaluated on scans of calibration bodies in two systems, a Hexapod CT [35] and the robot CT system shown in Figure 2. The sphere distances of the calibration objects and the corresponding errors were analyzed in relation to reference measurements and compared with the results of the offline calibration methods from [14,35], which showed comparable results.

#### 4.2.3. Conclusion to Image-Based Geometric Calibration

Image-based calibration enables highly accurate robotic CT scans with a wide range of CT setups, often requiring little to no additional hardware. Image-based calibration techniques have shown the potential to transform even simple 2D X-ray systems into fully functional 3D CT scanners, as demonstrated by [121].

However, image-based calibration approaches come with certain limitations. Offline calibration methods initially require a sufficiently high repeatability for the desired resolution. In addition, they always require a second CT scan, resulting in increased time and costs. Furthermore, offline calibration methods are only practical in the context of robotic CT systems if the object can be easily repositioned so that the calibration object can be scanned in the exact spatial location where the region of interest of the specimen was previously scanned. Offline calibration is therefore often impractical for large objects, such as automotive or aerospace components, and infeasible for stationary objects. In order to reduce this impractical problem, ref. [95] shows that local solutions can be used to provide data for other trajectories to calculate a better solution without the need for manual data acquisition. Additionally, previously acquired parameters can be used to refine a trajectory solution by incorporating X-ray projections from the new trajectory into the existing calibration dataset.

Since an initial scan of the calibration body is always required, the overall accuracy depends on the imaging chain and ultimately on the robot’s repeatability. Ref. [15] demonstrates that a resolution of approximately 75 µm voxel size can be achieved in comparative measurements of a sample with riveted joints using both a laboratory CT system and a robotic CT system.

Online calibration methods depend on the presence of sufficient features in the scanned object or require the strategic placement of markers that must be clearly identifiable and evenly distributed across the object. In practice, however, this can be challenging. For non-hollow objects, it may be impossible to evenly distribute markers, compromising calibration accuracy. Additionally, in dense objects, accurately detecting metal markers becomes difficult, further degrading the quality of the calibration. For medical purposes or less dense objects, these methods can be used very well and provide accurate results comparable to other state-of-the-art methods [122]. Online calibration methods that rely on prior information are often associated with high computational costs, as they use previously acquired 3D models of the sample and compare simulated forward projections with the recorded projections. Ref. [113] showed that results comparable to offline calibration can be achieved. Methods relying on data consistency conditions require the absence of significant artifacts and that the entire specimen is fully captured within the field of view of all projections, which limits the applicability to general robotic CT calibration scenarios. New approaches using differentiable forward projectors [136] or comprehensive AI [134] are very promising, but not yet widely used.

Currently, the literature lacks comprehensive evaluations and comparisons of calibration methods. It remains unclear which techniques are most suitable for specific objects and CT systems and how they affect the image quality and measurement uncertainties. Further studies are needed to robustly assess the performance and applicability of these methods for various object types and across various CT systems.

## 5. Challenge: CT Reconstruction

For some applications, two-dimensional projections are sufficient. For other applications, three-dimensional reconstructions are required to determine the attenuation coefficients of an object at specific points, along lines, within planes or as complete volumetric data for volume rendering. The reconstruction of a three-dimensional volume requires complex algorithms and computations as well as knowledge about the position of the X-ray source (point) and the position and orientation of the X-ray detector (frame) for each projection from the point of view of the object, hence the “pose”. Typically, as described in Section 4 about geometric calibration, more accurate knowledge about the pose results in a more accurate reconstruction. Figure 17 shows the exemplary measurement and reconstruction process of a LEGO model of a 1967th Ford Mustang, including a subset of projections after the measurement and a slice of the CT volume after the reconstruction. This model has a length of 34 cm, a width of 14 cm, a height of 10 cm and was measured with the Deggendorf Institute of Deggendorf twin robotic CT system using a circular trajectory as described in [14,137]. In this section, the relevant reconstruction methods and relevant challenges with regard to twin robotic CT systems are introduced.

### 5.1. Reconstruction Methods

In mathematics, one speaks of an inverse problem whenever one wants to determine the cause of an observed effect. In X-ray CT, the weakened intensity of radiation is measured along various rays through an object to determine the attenuation distribution within that object. Thus, CT is an inverse problem. Inverse problems are typically ill-posed in the sense of Hadamard [138], meaning there is no solution, or no unique solution, or no solution that depends linearly on the given data. Natterer shows in [139] that CT is not an exception to this rule. Regularization methods allow the computation of stable solutions for ill-posed inverse problems. An introduction to the regularization of ill-posed problems is provided, for example, by Louis in [140] or Rieder in [141]. Buzug [142] and Zeng [43] give a detailed introduction to the physical and mathematical backgrounds of CT reconstruction.

In the next sections, analytic reconstruction methods (Section 5.1.1), iterative reconstruction methods (Section 5.1.2) and deep learning reconstruction methods (Section 5.1.3) are introduced. Thereby, our focus is on the transformation from the projection domain to the reconstruction domain, as required for the main challenges regarding twin robotic CT systems in Section 5.2.

#### 5.1.1. Analytic Reconstruction Methods

Analytic reconstruction methods try to determine an attenuation function *f* that holds $P_{O}\left(G_{i},f\right)=I_{i}$ for given poses $G_{i}$ and projections $I_{i}$,i∈{1,…,s},s∈N, by finding an inversion formula of $P_{O}$. In the case of cone-beam CT, the projection map $P_{O}$ is called X-ray transform and is defined by$$I\left(w,h\right)=P_{O}\left(G,f\right)\left(\theta_{w,h}\right):=\int_{0}^{\infty }f\left(\phi \left(G\right)+t\theta_{w,h}\right)\;dt,$$
where I(w,h) is a pixel of the projection image I indicated by the row index w∈N and the column index h∈N, $\phi :R^{j}\to R^{3}$ maps the pose G one-to-one to a coordinate in $R^{3}$ that represents the focal spot of the X-ray tube, and $\theta_{w,h}\in \{x\in R^{3}:∥x∥=1\}$ is an unit vector that points from the coordinate of the focal spot of the X-ray tube to the pixel I(w,h).

The most commonly used analytic reconstruction method is the Filtered Back Projection (FBP). In this method, each projection is first filtered with a suitable kernel. Then, in simple terms, each point of the filtered projection is back-projected along the line connecting this point to the position of the focal spot of the X-ray tube.

An alternative reconstruction method is called filtered layergram, where the steps of filtered back projection are reversed; first, back projection is performed, and then filtering is applied in the reconstruction domain. Compared to filtered back projection, which can be implemented very efficiently, filtered layergram methods are computationally intensive and therefore much less common [142].

As the 3D case is particularly relevant in practice, the inversion of the X-ray transform has primarily been studied for the 3D case. Tuy shows in [48] a formula for the inversion of the 3D X-ray transform. Although this does not lead directly to an algorithm, Tuy demonstrates that a stable inversion is possible if the corresponding curve or trajectory of the X-ray source satisfies the Tuy condition named after him. The Tuy condition intuitively means that every plane containing a point of the object to be reconstructed must intersect the curve or trajectory of the X-ray source at least at one point. The number of intersection points is characterized by a function called Crofton symbol. Formulas for inverting the 3D X-ray transform in the presence of a Tuy curve and derivable reconstruction methods have already been thoroughly investigated—see, e.g., Smith [143], Grangeat [144], Defrise and Clack [125] and Kudo and Saito [145]. The Crofton symbol, which typically represents a discontinuous function, appears in all mentioned approaches and its generalized derivative must be determined. This often leads to special numerical treatments.

Katsevich provides in [146] another formula for inverting the 3D X-ray transform for arbitrary Tuy curves and formulates a reconstruction algorithm. Katsevich employs a suitable weighting function to avoid the derivative of the Crofton symbol, resulting in a stable reconstruction method. The determination of the weighting function is numerically complex for arbitrary Tuy curves. Another inversion formula for the 3D X-ray transform in the case of arbitrary Tuy curves is presented by Louis in [147]. Louis’s approach is similar to that of Defrise and Clack [125], but introduces new operators to achieve a suitable structure of the inversion formula. Since Louis’s inversion formula is based on the adjoint X-ray transform, the approximate inverse [148] serves as a regularization method to develop stable reconstruction methods. Oeckl shows in [149] a formula for inverting the X-ray transform that holds for arbitrary Tuy curves and for any dimension. The Crofton symbol does not need to be derived in this approach. This approach can be used with both methods, filtered back projection and filtered layergram.

In 1984, Feldkamp, Davis and Kress (FDK) [150] presented an approach that is not based on an inversion formula for the 3D X-ray transform and was developed only for a circular trajectory by adding suitable weights to both the filtering and back projection operations. Nevertheless, the FDK algorithm evolved to the standard algorithm for cone-beam CT because of its simplicity and efficiency. Since a circular trajectory does not satisfy the Tuy condition in the 3D case, artifacts arise in the reconstruction.

Besides circular trajectories, two additional important scanning trajectories are commonly used in analytic reconstruction: helical (spiral) CT, which collects a Tuy-complete dataset [151,152], and laminography/tomosynthesis, which collects a Tuy-incomplete dataset [153,154]. Compared to circular CT, helical/spiral CT provides Tuy-complete data not only in the central slice but throughout the entire object, making it particularly suitable for scanning long (axial) objects. In contrast, laminography/tomosynthesis does not achieve Tuy-completeness in the central slice or in other slices. However, it is well suited for imaging large, thin (transverse) objects.

Practically, while FBP is computationally efficient, it does not allow for incorporating boundary conditions that arise, e.g., from polychromatic beam hardening, noise, arbitrary trajectories or coarse or non-equidistant/non-equiangular sampling.

#### 5.1.2. Iterative Reconstruction Methods

Iterative reconstruction methods try to determine a discrete volume *V* that satisfies $AV=I_{i}$ for a given system matrix *A* and projections $I_{i}$,i∈{1,…,s},s∈N, by finding an optimization formula $V^{t+1}=X\left(A,V^{t},I_{i}\right)$ for minimizing an error term, e.g., the reprojection error $∥AV-I_{i}∥$, with *A* as a discretization of the projection map $P_{O}$ for given poses $G_{i}$, *t* as the iteration index and $V^{0}$ as an initial guess.

One of the earliest and simplest iterative reconstruction methods is the class of Algebraic Reconstruction Methods (ARMs), which will be discussed in more detail: In 1937, Kaczmarz [155] introduced an iterative method for solving systems of linear equations, which was later rediscovered and applied in 1970 by Gordon et al. [156], the Algebraic Reconstruction Technique (ART) was discussed in 1972 by Gilbert [157], resulting in the Simultaneous Iterative Reconstruction Technique (SIRT), while also explored in 1984 by Andersen [158], leading to the Simultaneous Algebraic Reconstruction Technique (SART). This method can be implemented in three key steps: computing a forward projection to simulate the projection, calculating the difference between the measured and simulated projections, and applying a back projection operation to update the volume voxels. These three steps are repeated until a termination criterion is met.

The key difference between ART, SART and SIRT lies in how the volume update is performed: ART updates the volume after processing each individual ray, SART after each projection and SIRT after all projections in a full iteration. Accordingly, there are also differences in the convergency and parallelizability. In 2003, Jiang and Wang [159] studied the convergency of some iterative algorithms and Wang and Jiang [160] combined SART and SIRT into the Ordered-Subset Simultaneous Algebraic Reconstruction Technique (OS-SART) with a block-iterative approach [161] to update the volume after processing not only one projection but multiple projections. This allows a more efficient implementation than SART or SIRT in memory hierarchies including GPUs. OS-SART is identical to SART and SIRT if the block size is set to one or to all projections, respectively.

Other iterative reconstruction methods are also based on the forward projection and back projection of rays and projections: In 1993, Sauer and Bouman [162] solved the equation system with the Gauss–Seidel method. Gradient-based methods like Steepest Descent or Conjugate Gradient Least Squares (CGLS) have a better step size and a better convergency but need more memory. Practically, due to limited data, low-dose and noise, there is not a unique solution but multiple solutions and penalty or regularization terms are used to obtain not any but the “best” solution dependent on the use case. Therefore, regularizations and statistical/model-based methods make use of a priori information about the measurement system and/or the measured object. Examples of regularizations and statistical/model-based methods will be given in Section 5.2.3 under “Limited Data”.

In 2008, for further reading, Wang et al. [163] gave a review and outlook on X-ray CT research and development emphasizing analytic reconstruction, iterative reconstruction, local/interior reconstruction, flat-panel based CT, dual-source CT, multi-source CT, novel scanning modes, energy-sensitive CT, nano-CT, artifact reduction, modality fusion and phase-contrast CT.

#### 5.1.3. Deep Learning Reconstruction Methods

Deep learning may be used for preprocessing the projections or sinograms before the reconstruction, for postprocessing the volume after the reconstruction or for the reconstruction as domain transform itself, either by embedding deep learning operators into analytic or iterative reconstruction methods or vice versa by embedding deterministic and known operators into deep learning reconstruction methods. Like analytical and iterative reconstruction methods in recent decades, deep learning reconstruction methods were initially developed in two dimensions and have been extended to three dimensions in recent years.

In the following, some review papers about deep learning reconstruction are summarized: In 2018, Wang et al. [164] saw image reconstruction as a new frontier of machine learning and gave a good summary of the research up to then. In 2020, Zhang and Dong [165] published a review on deep learning in medical image reconstruction. They summed up the most important reconstruction methods and neural networks with the relevant mathematical background. Further, they substantiated more end-to-end deep learning models including not only the reconstruction but also the image analysis, like detection, classification and segmentation. Finally, they saw a challenge in the availability of sufficient labeled data in general and for specific tasks. In 2020, Wang et al. [166] published a review on deep learning for tomographic image reconstruction. They summed up four challenges: (1) big data, weakly supervised and unsupervised learning, i.e., the lack of annotated and labeled training data or training data at all; (2) point of care (POC), hybrid and autonomous imaging; (3) network stability and image quality assessment; and (4) explainable/interpretable and ethical artificial intelligence (AI). In 2024, Bellens et al. [167] published an overview and qualitative examples of machine learning and deep learning methods in industrial X-ray computed tomography sorted along the data processing workflow, including the following: scan parameter optimization, viewpoint optimization, X-ray deblurring, sinogram completion, reconstruction, segmentation, reconstruction deblurring, noise reduction, beam hardening correction, scatter correction, ring artifact correction, metal artifact reduction, super resolution, general reconstruction enhancement, reconstruction denoising and others. That is, deep learning methods are involved from the planning of the scan to the final reconstruction or even classification, anomaly and defect detection.

Examples of deep learning approaches for preprocessing or postprocessing will be given in Section 5.2.3 in “Limited Data”. Examples of deep learning approaches for the domain transform itself will be given here:

In 2018, Zhu et al. [168] presented a framework for image reconstruction with automated transform by manifold approximation (AUTOMAP) and demonstrated results for magnetic resonance imaging. AUTOMAP directly learns the mapping between the measurement and the reconstruction with a fully connected first layer. Basically, this approach is possible not only for magnetic resonance imaging but also for computed tomography.

In 2019, Syben et al. [169] implemented PYRO-NN as a framework to embed deterministic and known linear operators like forward/back projection into deep learning neural networks. In AUTOMAP, the full signal pipeline is learned, i.e., from the input measurement to the output reconstruction including the system matrix. With PYRO-NN, the full signal pipeline is modeled with the deterministic and known reconstruction operators, decreasing the number of parameters and thus the memory and the amount of necessary training data for the deep learning.

In 2020, Lagerwerf et al. [170] extended the FDK algorithm to the Neural Network Feldkamp–Davis–Kress algorithm (NN-FDK). This method for a circular cone-beam geometries learns a set of FDK filters and combines the FDK reconstructions achieved with these filters and improved reconstruction quality in case of high-noise, sparse angles and large cone angles.

In 2022, Rückert et al. [134] implemented neural adaptive tomography (NeAT) combined with a three-dimensional reconstruction approach, an octree-based hierarchical data structure, handling of sparse view and limited angle and geometric image-based online calibration as described in Section 4.2.2. They observed that, in case of the neural adaptive tomography, the neural network hallucination artifacts look physically plausible and are therefore harder to distinguish from a correct reconstruction.

In 2022, Fu and de Man [171] focused on domain transforms and its hierarchical decomposition. Domain transforms in CT cover the high-dimensional mapping from the projection space to the reconstruction space or vice versa. Due to this high-dimensionality, approaches like AUTOMAP are only implemented for small images. Fu et al. make use of the sparsity of the transform and the fact that zero-weights do not have to be trained and stored. Their approach was tested by them for the case of a two-dimensional circular CT but can be extended for the case of a three-dimensional circular CT, too.

In 2024, Ye et al. [172] employed a differentiable reconstruction for arbitrary CBCT orbits (DRACO), a shift-variant FBP algorithm optimized for arbitrary trajectories through a deep learning approach that adapts to a specific trajectory. In [125], the redundancy weights are computed analytically. In [172], the redundancy weights are learned. Results were shown for a sinusoidal orbit, a circle plus arc trajectory and random trajectory points on a sphere.

A challenge for deep learning reconstruction methods is the gathering of annotated and labeled training data. Both the measurement of the real data and the annotation and labeling of these data are associated with high efforts and costs. Before refining a model with expensive real data, simulated projections may help for prototyping, research and development. One advantage is the reproducibility of the simulation and the ability to simulate edge cases that are rare or difficult to measure in reality. Another advantage is the avoidance of manual annotation and labeling as the ground truth is completely known. In the following, some X-ray simulation use cases and frameworks are given: In 2022, Vienne et al. [173] assessed the influence of CT acquisition parameters on flaw detectability through simulation using CIVA. In 2023, Sukowski et al. [174] automated 3D defect detection based on simulated reference using XSimulation. In 2024, Fleßner et al. [175] analyzed image data to detect a CT system’s error state and identify the corresponding root cause using aRTist. In 2024, Wu et al. [176] published an overview of the open-source CT simulation environment XCIST.

### 5.2. Challenges Regarding Twin Robotic CT Systems

One of the main advantages of a twin robotic CT system is its degrees of freedom, which enable the system to carry out arbitrary trajectories and a local tomography of regions of interest (ROIs) [12,15,177]. However, the new capabilities regarding the hardware result in new challenges and requirements regarding the software and the reconstruction methods:

In the case of the circular cone beam-computed tomography of the 1967th Ford Mustang in Figure 17, differences between the reconstruction results of the cropped FBP and cropped SART are hard to spot. However, the projections were measured to only cover the car front with the car engine as region of interest. Figure 18 shows the complete reconstruction of the complete object with an FBP and SART. For both reconstruction methods, the front of the car looks fine while the rear of the car shows limited-angle artifacts. The edge of the projections show up as a circle or ellipse (the rotation axis and reconstruction slice are not perpendicular) with a strong intensity in the FBP and a weak intensity in the SART reconstruction. Corrections, which will be introduced in Section 5.2.2, were not applied. In the case of arbitrary trajectories and region of interest reconstruction using all the degrees of freedom of the twin robotic CT system, the differences between analytic and iterative reconstruction methods would be much more significant.

Both arbitrary trajectories and region of interest reconstruction can be seen as the main challenges for the reconstruction in a twin robotic CT system. For analytic and iterative methods, arbitrary trajectories and region of interest reconstruction will be focused on in more detail in Section 5.2.1 and Section 5.2.2, respectively. For deep learning reconstruction methods, the capabilities depend on the architecture of the network and the selection of its operators: NeAT [134], for example, is capable of arbitrary trajectories but not region of interest reconstruction. AUTOMAP [168], for example, as a fully connected network, may be capable of both arbitrary trajectories and region of interest reconstruction but at the cost of impractical memory consumption for the parameters and impractical computation time for the training.

#### 5.2.1. Arbitrary Trajectories

Although the term “Arbitrary Trajectories” has been used in the previous sections, it requires a more detailed definition for the reconstruction. “Arbitrary Trajectories” refer, first, to arbitrary curves (for example, in opposite to circular or helical curves), second, to arbitrary sample points (for example, in opposite to equidistant or equiangular sample points) and, third, to non-constant transformations from the X-ray source to the X-ray detector (for example, in opposite to a C-arm system).

**Arbitrary Trajectories using Analytic Reconstruction Methods:**
In all analytic reconstruction methods for trajectories that satisfy the Tuy condition, the Crofton symbol must be computed. This requires determining the number of intersection points of all planes through all reconstruction points with the trajectory. This is feasible for well-behaved trajectories that exhibit certain symmetries, such as a helical trajectory, as shown by Weber [178]. For arbitrary trajectories, this can become computationally intensive.

Many approaches for CT reconstruction algorithms for trajectories that satisfy the Tuy condition still yield a reconstruction algorithm even when the Tuy conditions are not fulfilled, as seen in Schön [179]. The Crofton symbol must still be determined or at least suitably chosen in this case. In some instances, calculating the Crofton symbol may be simplified; however, in every case, reconstruction artifacts will arise due to the lack of measurement data.

When a CT scan does not sample the trajectory with numerous measurement points, but instead consists of only a few projections from various directions, analytical reconstruction methods reach their limits in the form of the occurrence of artifacts. This is because analytical methods model a CT scan with an integral transformation, and this model is then too far from reality. In 1997, Noo et al. [180] presented a possible approach for a cone beam reconstruction from general discrete source positions using binning and interpolation in the projection domain. In 1994, Defrise and Clack [125] used a shift-variant filtering with the redundancy weights computed analytically. In 2024, Ye et al. [172] used a shift-variant filtering with the redundancy weights learned. The latter supports trajectories where it is too complex or not possible to formulate or compute an analytical formula. In 2022, Russ et al. [181] presented a fast reconstruction scheme for arbitrary acquisition orbits based on filtered layergram, where the filter is not derived from the inversion formula but must be heuristically determined.

**Arbitrary Trajectories using Iterative Reconstruction Methods:** The algebraic reconstruction methods, gradient-based reconstruction methods and statistical/model-based reconstruction methods are based on the forward projection and the back projection operators. These operators only depend on the actual projection and not on the overall orbit or trajectory. That is, iterative reconstruction methods are capable of arbitrary trajectories.

#### 5.2.2. Region of Interest Reconstruction

In region of interest reconstruction, a distinction must be made between complete (non-truncated) and truncated data. According to Clackdoyle and Defrise in [182], data are complete or non-truncated for a projection, i.e., a specific position of source and detector, if the raysum of all rays through the source is either measured or known to be zero. Incomplete projections have the unmeasured rays at arbitrary pixels. Truncated projections have the unmeasured rays at the extremities of the nonzero pixels. That is, truncated projections are a subset of incomplete projections. According to Clackdoyle and Defrise, the case of full angular coverage but with all projections truncated on both sides is called the interior problem [25,183]. Mathematically, its solution is not unique.

Figure 19 shows an out-of-the-box region of interest reconstruction with FBP and SART. Here, not the complete object but only the region of interest was reconstructed. The FBP, as an example of an analytic reconstruction, is clear. The SART, as an example of an iterative reconstruction, contains artifacts. In the following, the reasons for these differences will be explained.

**Region of Interest Reconstruction using Analytic Reconstruction Methods:** For analytic reconstruction methods, single voxels of the reconstruction volume can be computed independently. In the case of non-truncated data, the object projections are covered by the detector with only zero raysums at the edge of the projection. A region of interest reconstruction is possible without the occurrence of artifacts. In the case of truncated or partially truncated data, the objects projections intersect the edge of the detector also with non-zero raysums at the edge of the projection as shown in Figure 17. Without a correction algorithm, this edge will lead to an edge in the reconstruction as seen in Figure 18.

To overcome the interior problem, Li et al. [131] use a small detector for the region of interest and a large detector for the background correction for the filtering. There are other approaches using only one scan and decomposing the filter into two steps where the first step consists of a local operation. For example, Noo et al. [184] decompose the filter into a derivative and an inverse Hilbert transform and Xia et al. [185] show a decomposition of the filter into a 3D Laplace filter and a 3D residual filter.

**Region of Interest Reconstruction using Iterative Reconstruction Methods:** In iterative reconstruction, the entire system of equations must be solved to minimize artifacts, as shown in Figure 19. This directs to the straightforward approach for region of interest reconstruction: a high-resolution reconstruction of the entire object is first computed, incurring significant memory and runtime costs, and then cropped to the desired region. More advanced techniques reduce memory and runtime costs by lowering the resolution outside the region of interest, utilizing adaptive or multiple grids (“multigrid”). Initially, these methods were motivated by the unknown dimensions and positions of the object of interest, particularly when dealing with fluids and gases observed through optical systems [186]. Later, these methods were adopted for X-ray imaging systems [187]. A key challenge is determining the appropriate grid resolutions: higher resolutions are more costly, while lower resolutions can introduce artifacts. This choice depends on the object being imaged: Regions with small variations or blurry objects like fluids and gases can be reconstructed with a lower resolution to reduce the costs. Regions with large variations or sharp-edged objects like industrial parts must be reconstructed with a higher resolution to reduce the artifacts. Other difficulties include managing transitions between resolutions, handling interpolation artifacts and avoiding moiré or aliasing artifacts caused by varying resolutions across different parts of the volume. In 2012, Gregson et al. [188] replaced the reconstruction volume grid with sample points and radial basis functions allowing a smooth transition between various resolutions and several regions. In 2015, Kopp et al. [189] computed the FBP of the complete object, zeroed out the region of interest and subtracted the forward projections from the measured projections, resulting in projections of only the region of interest. In 2024, Jung et al. [137] avoided multiple grids by moving the residuals outside the region of interest from volume to projection space. These residual projections are meant to cover equation system inconsistencies and thus may help not only with region of interest reconstruction but also in other applications and use cases like artifact reduction.

#### 5.2.3. Other Challenges

While arbitrary trajectories and region of interest reconstruction are the main challenges in twin robotic CT reconstruction, there are other challenges, too: The first challenge is data volume, impacting both storage requirements and computational runtime. The second challenge is limited data, including data incompleteness, noise and artifacts, particularly when scanning large or complex-shaped objects.

**Data Amount:** Standard system configurations of industrial CT scanners include a detector matrix of 3072 × 3072 pixels, resulting in approximately 18 MiB per projection (assuming uint16 format). For a typical circular trajectory with 2400 projections, the total projection data amount to about 45 GB, and the reconstructed volume (in float32) can reach 90 GB.

Twin robotic CT systems, however, support larger object sizes and allow for more flexible scan trajectories with a significantly higher number of projections. For example, in a scan of a BMW 4 Series Gran Coupé door, 14,400 projections at 3072 × 3072 pixels were acquired and later stitched into 800 high-resolution projections of 7119 × 13,179 pixels, resulting in approximately 250 GB of projection data alone. To handle this data volume, the complete reconstruction pipeline is optimized from the reconstruction mathematics and the reconstruction implementation to the dedicated GPU-Servers. Both iterative and deep learning-based reconstruction methods may use analytic components either for the reconstruction itself or for a precomputation. For example, in [190], the volume for the algebraic reconstruction is not initialized with zeroes but with the result of a Filtered Back Projection. Approximations of the mathematical model help to reduce the computation time, for example, the unmatching pairs of the forward projection and back projection operators [191,192,193]. Hierarchical structures, for example, different resolution levels like multigrid [186] or different compression levels like Wavelet [194,195], help to reduce the memory consumption. Typically, the output attenuation coefficients are stored in a voxel grid. But there are also other representations: In 2012, Gregson et al. [188] replaced the reconstruction voxel grid with sample points and radial basis functions. In 2022, Rückert et al. [134] replaced the reconstruction voxel grid with a neural implicit representation. A technical and performance survey is given in [196].

**Limited Data:** The term limited data contains some conditions under which a reconstruction is further complicated: truncated data, limited angle and sparse view. Here, also low-dose and noise are included. The truncated data and interior problem has been discussed in Section 5.2.2. Ideally, both the Tuy condition [48], ensuring sufficient geometric coverage, and the Nyquist–Shannon condition [142], preventing aliasing, are met. The Tuy condition is violated in case of limited angle. The Nyquist–Shannon condition is violated in case of a sparse view, i.e., an insufficient number of projections compared to the number of projection pixels and volume voxels.

A twin robotic CT system is, on the one hand, capable of large object sizes but, on the other hand, limited regarding the X-ray components weights, e.g., the X-ray tube weight and power. Thus, compared to systems with more powerful tubes, a twin robotic CT system is more suitable for large thin objects (like bikes) or large holey objects (like cars) and more sensitive to objects with strong attenuation. In particular, metal parts of the object itself or of the calibration phantom will cause beam hardening and beam hardening artifacts. Even if both conditions are met, strong attenuation effects (e.g., beam hardening) can still result in insufficient data for certain regions or voxels. An optimized trajectory can reduce but not entirely eliminate strongly attenuated rays. Consequently, not only must the trajectory be optimized to maximize high-quality data, but the reconstruction process must also be adapted to minimize the influence of low-quality data. Therefore, rays with strong attenuation can be down-weighted or ignored [21,197].

In many cases, noise and artifacts can be reduced using additional a priori information about the measurement system and/or the scanned object. The following are some examples:

Information about the system noise characteristics: In 1977, Rockmore and Macovski [198] created a statistical model of the measurement system noise characteristics and the underlying Poisson process and used a maximum likelihood approach for the reconstruction. This approach was followed by other statistical/model-based iterative reconstruction methods, for example, Penalized Weighted Least Squares (PWLS) [199] and derivatives [200,201,202]. Statistical/model-based iterative reconstruction methods can account not only for the system noise characteristics but also for other a priori information about the measurement system and/or the measured object, as discussed in the following paragraphs. In 2017, Syben et al. [203] learned the discrete optimal reconstruction filter for the filtered back projection directly from the continuous Ramp filter. For data without noise, their result was similar to the Ram–Lak filter. For data with noise, their approach has the potential to learn an optimal discrete filter for the given noise characteristics. In the authors’ view, these observations may help in understanding deep learning and traditional analytic techniques such as Wiener filtering and discretization theory. In 2020, Hendriksen et al. [204] proposed Noise2Inverse, a self-supervised and Mixed-Scale Dense-based method [205] for denoising linear image reconstructions. They made use of the fact that the noise of voxels is not independent, as assumed by other methods, but dependent as of the back-projection of the reconstruction.

Information about the object signal characteristics: Methods with compressed sensing [206] like total variation minimization [207] or wavelet compression [208] exploit the sparsity of natural signals by assuming that most structures in an image exhibit minimal variation. This allows for high-quality reconstructions with fewer projections and reduced noise. In 2015, Schön et al. [209] reduced the cycle time in process-integrated computed tomography using compressed sensing. In 2017, Roemer et al. [210] computed a differential SART using the difference of the actual measurement and a reference measurement of an object, similar in shape and orientation. Their approach makes use of the sparsity of the difference projections and the simplicity of the total variation minimization in the difference domain and considers misalignment between the actual object and the reference object.

Information about the object material composition: Practically, the Beer–Lambert law is used to convert from measured intensities to linear raysums. This process is also known as log-correction and covers mono-energetic X-rays and mono-material objects. In the real case of poly-energetic X-rays, one can observe beam hardening in the measurement and beam hardening artifacts in the reconstruction caused by this approximation. In case of mono-material objects, a characteristic line for beam hardening correction can be computed or measured. In case of multi-material objects, the forward projection can be extended from a mono-energetic model to a poly-energetic model like pSART [211]. pSART is based on knowledge of the object materials and the X-ray spectrum and computes the attenuation coefficients for a given reference energy.

Information about the object material discreteness: In non-destructive testing, there are many use cases requiring a segmentation. Typically, first, a reconstruction is computed and, second, the reconstruction is segmented. But there are also approaches that perform the reconstruction and the segmentation simultaneously: Originally motivated by the detection of the absence or presence of atoms in a crystal in electron microscopy, the binary or discrete reconstruction received more attention [212,213]. DART [214] was able to improve the reconstruction resolution for both simulated data [214] and measured data like bones [215]. With DART, by using the information about the object material discreteness, it was possible to either increase the reconstruction resolution with the same projections or to decrease the X-ray dose with the same reconstruction resolution. In each iteration, DART segments the voxels into fixed and unfixed voxels while updating only the unfixed voxels. Thereby, voxels are considered fixed if there is a high probability of coincidence between the voxels’ attenuation coefficient and the truth and if voxels are not at a boundary between materials. In 2012, Vlasov et al. [216] proposed a similar approach for artifact preventive reconstruction in few-view computed tomography not with additive but with multiplicative algebraic reconstruction. In 2017, SART-FISTA-TV and DART-FISTA-TV were used for limited-angle trajectories applied to robotic inspection [13,217]. These two proposed algorithms combine Total Variation (TV) regularization and the Fast Iterative Shrinkage-Thresholding Algorithm (FISTA) to increase the convergence speed. In 2019, Poly-DART [218] combined DART [214] and pSART [211]. This combination improves DART with the poly-energetic forward projection model or pSART with the knowledge of discrete material steps. While DART segments the voxels only based on the voxels’ attenuation coefficient, Poly-DART extends this segmentation with some random seed points. In Tabu-DART [219], the segmentation was achieved with a probability map.

Information about the object shape: The object shape can be learned or retrieved, for example, from a CAD model [220] or from other sensors in case of a multimodal setup. Laurentini et al. [221] computed the visual hull to separate air and object. This can be used for optical and X-ray data: a voxel can be considered an air voxel if there is at least one ray through this voxel without attenuation. In 2013, Schrapp et al. Schrapp et al. [222] additionally used ultrasound data to support the X-ray reconstruction by detecting and providing the inner object surfaces and edges. In 2012, Stayman et al. [223] modeled and optimized as part of the “known-component reconstruction” (KCR) the transformation of known components like pedicle screws. In 2012, Xu et al. [224] combined statistical iterative reconstruction and dictionary learning as a sparse representation of the object. Therefore, a dictionary of image patches or atoms was learned from normal-dose reconstructions and used for the improvement of low-dose reconstructions of the same or a similar object. In 2018, Zheng et al. [225] extended the PWLS reconstruction method with a regularization based on a Union of Learned TRAnsforms (ULTRA) to PWLS-ULTRA. Similar to [224], this union of transforms was pre-learned from image patches extracted from reconstructions of the same or a similar object. In contrast to [224], the approach of [225] is not only for two-dimensional slices but also for three-dimensional volumes. In 2018, Pelt et al. [205] improved a filtered back projection reconstruction from limited data using Mixed-Scale Dense convolutional neural networks. In comparison to encoder–decoder convolutional neural networks, Mixed-Scale Dense networks use the images not only of the previous layer but also of all previous layers. The approach learns characteristics of objects and was benchmarked against limited number of projections, limited exposure time, limited angular range and also limited quality of training images. Due to the available frameworks, the implementation was only for two-dimensional slices and not for three-dimensional volumes.

### 5.3. Conclusion to CT Reconstruction for Robot CT Systems

In the state of the art, arbitrary trajectories are challenging for analytic reconstruction and region of interest reconstructions are challenging for iterative reconstruction. However, as shown, there are promising approaches in both directions, making analytic reconstruction methods capable of arbitrary trajectories and iterative reconstruction methods capable of region of interest reconstruction. The combination of arbitrary trajectories and region of interest reconstruction is already feasible but comes with increased computational and memory demands. This is a general challenge in robotic CT systems, which offer greater flexibility but also deal with larger objects and often significantly higher data volumes than standard CT systems. Deep learning may help not only to reduce the computation and memory costs in comparison to complex analytic or iterative methods but also with other challenges like limited data. Ultimately, there is no universal solution, but rather a diverse set of analytic, iterative and deeplearning-based methods that can be combined to suit specific applications.

## 6. Conclusions

Robotic CT is an exceptionally versatile and powerful tool for metrology and digitization. However, the detailed challenges currently often limit its efficient application. A key limitation lies in the lack of comprehensive evaluations and testing. This complicates determining whether robotic CT can effectively address the described challenges with the currently available methods for each specific scenario. The reliable use of robotic CT therefore requires experts and incurs relatively high costs. However, as the methods outlined here continue to develop, these costs and dependencies on expertise are expected to decrease over time.

Several research groups are actively addressing these challenges. Once robotic CT systems are capable of autonomously generating and following optimized scanning trajectories (Section 3: CT Trajectory Optimization), achieving high precision in geometric calibration (Section 4: Geometric Calibration), and reliably extracting and reconstructing the measured data (Section 5: CT Reconstruction), we believe robotic CT will revolutionize NDT. Robotic CT will unlock new possibilities for CT scanning, enhancing non-destructive testing in established industries and enabling applications in entirely new environments.

The technology is currently at a stage comparable to the early years following the introduction of industrial CT scanners—still developing but already a powerful tool. Realized systems have been successfully used in various but still isolated application scenarios. These systems are particularly effective for analyzing large or complex geometries. With the global trend towards electric vehicle development, there are significant opportunities for utilizing such systems, for example, efficiently analyzing an entire battery pack in a single CT scan while enabling high-resolution scans for specific regions where necessary.

Another example is the inspection of large casting or giga-casting components. For these complex and time-consuming tasks, twin robotic systems enable a practical two-stage inspection workflow: First, fast 2D radiographs are acquired to identify potentially critical areas or anomalies. Then, 3D CT is selectively applied to those regions where additional detail is needed for verification or defect characterization. While this approach can significantly reduce scan time, it may also lead to motion blur and decreased image quality in some cases. Therefore, new trajectory optimization and reconstruction methods must be developed, specifically designed for smooth, high-speed CT scanning, while mitigating motion-induced artifacts.

One of the strengths of industrial robots is their repeatability during positioning, as demonstrated hour after hour in many production lines for repetitive processes. Developments in new X-ray sources and X-ray detectors as well as innovative developments in the field of industrial robots enable a significant increase in accuracy and resolution in imaging. We are currently achieving a long-term stable and reliable resolution of around 75 µm voxel edge length with robotic CT systems. This high precision is more than sufficient for many applications in quality control and metrology and opens up new possibilities for the combination of large-scale and detailed analyses.

## 7. Outlook

In the following, we outline several exemplary and particularly promising areas for further improvement.

**Mobility/Flexibility**: Different applications require varying degrees of flexibility. If the calibration challenges discussed in Section 4 can be resolved to the extent that application-specific measurements can be reliably generated even with less stable manipulators, numerous new scanning scenarios may become feasible. Some applications might require more agile robots with additional or specialized joints. Other applications might require very small, flexible manipulator systems that can be freely placed inside hollow objects such as car bodies. Other scenarios might require freely movable X-ray components, for example, by mounting them on mobile platforms or even drones to scan large infrastructure. As an example, Figure 20 shows a potential new application that might demand enhanced mobility: non-destructive testing for infrastructure [226].

**Figure 20 sensors-25-03076-f020:**
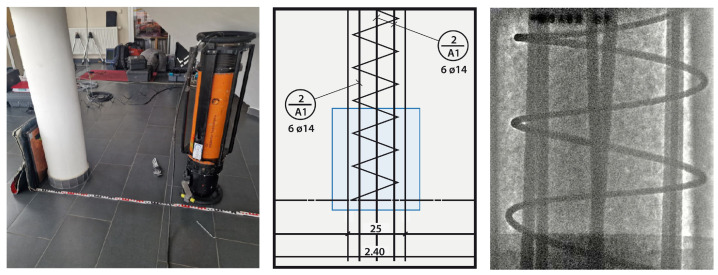
Example from bauray GmbH for verifying the reinforcement content of a column Schulze [226]: (**Left**) Photo of the column with an X-ray source and a detector. (**Center**) Construction plan. (**Right**) X-ray image. Reprinted with permission.

**Scanning unknown objects**: If there is no digital prior knowledge about the object, experts are currently required to manually plan the CT trajectory. To ensure a safe CT scan without robot collisions, sensors can be employed to map the object’s surface to define collision boundaries. The automatic trajectory optimization methods described in Section 3 rely on information of the object’s geometry and internal structure. As such, these current methods are not applicable without prior information. Zaech et al. [45] have developed a method that uses data from already acquired projections for real-time trajectory optimization, reducing the need for prior information in medical applications. To enable industrial CT scans without sufficient prior knowledge, comparable methods need to be developed and evaluated across various use cases.

**Scanning Speed**: Current robotic CT scans are time-consuming. This is partly due to suboptimal robot path planning and the intentionally slow movement of robots, which increases safety and minimizes acceleration forces on the X-ray components. Additionally, current robotic CT systems stop before generating each projection. To increase the speed of robotic CT systems, we expect future systems to continuously move the X-ray components along smooth trajectories during the scan: a technique known as fly-by scanning. To fully realize these benefits, new trajectory optimization methods must be developed, specifically designed for smooth, high-speed CT scanning.

**Reconstruction and multimodal imaging**: Robotic CT systems enable scanning of regions of interests within large objects. However, as mentioned in Section 5, due to the shape, size and density of many interesting use cases, it is often not possible to acquire sufficient information for reliable and precise reconstruction. On the one hand, this lack of information could be addressed using optimized reconstruction strategies and AI methods leveraging prior knowledge. On the other hand, additional information could be obtained by integrating supplementary sensor technologies, such as ultrasound [222], terahertz, infrared, radar or X-ray backscatter. In principle, these sensors could be mounted on the existing robots of the robotic CT system. Future research may focus on developing application-specific workflows to plan multimodal scanning trajectories and fusion methods to automatically generate the necessary data and extract the relevant information.

The increased flexibility of CT systems opens up entirely new possibilities for non-destructive imaging. Achieving this will not only require improved hardware and software, but also novel workflows and imaging strategies.

## Figures and Tables

**Figure 1 sensors-25-03076-f001:**
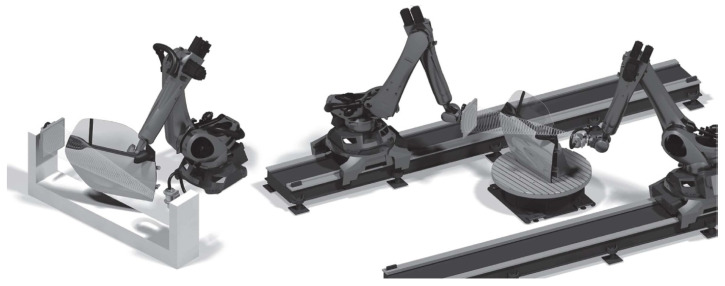
Sketch of two setups of industrial robotic CT systems scanning a car door [11]. (**Left**) Mono robotic CT system with one robot moving the object. (**Right**) Twin robotic CT system with two robots individually moving the source and the detector around the object.

**Figure 2 sensors-25-03076-f002:**
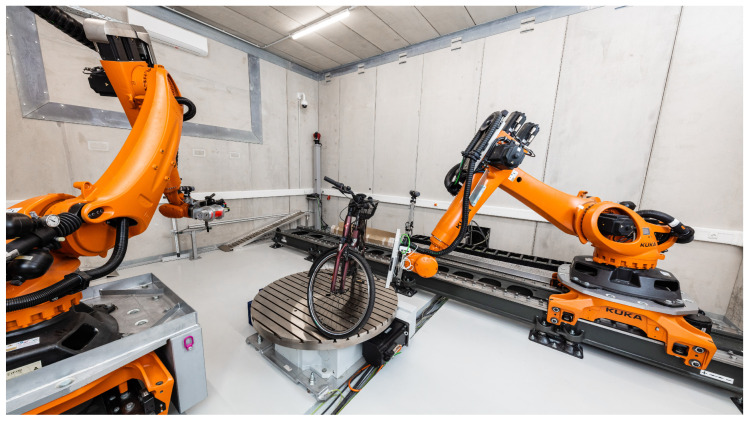
Twin robotic CT system at the Deggendorf Institute of Technology scanning a bike.

**Figure 3 sensors-25-03076-f003:**
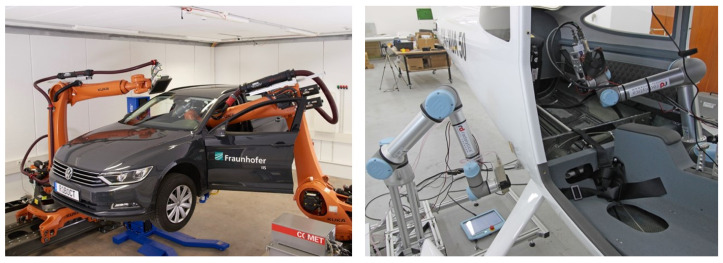
Robotic CT systems scanning regions of interest in large industrial components. (**Left**) Robotic CT system of Fraunhofer EZRT inspecting a car [7]. (**Right**) Mobile robotic CT system by Radalytica scanning an airplane [17]. Images printed with permission.

**Figure 4 sensors-25-03076-f004:**
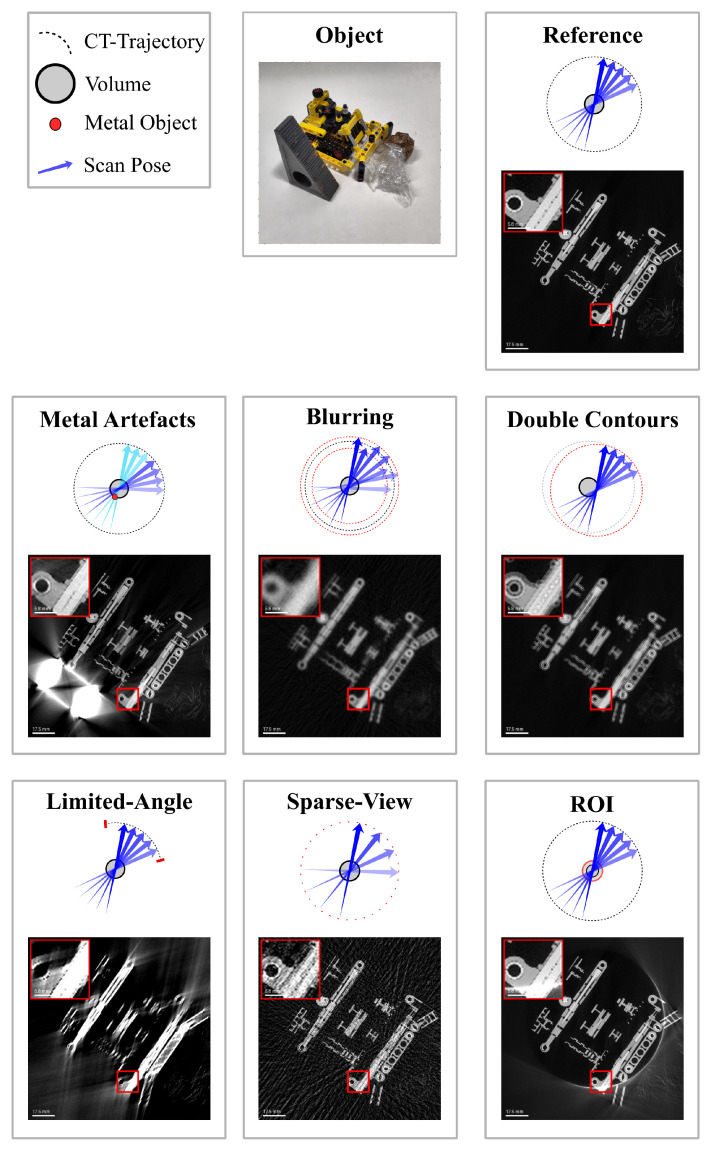
Overview of the most common CT artifacts in the context of RoboCT. All reconstructions were performed using an FBP algorithm. The data were acquired using the RoboCT system in Deggendorf. The specimen was a small Lego object, selected to independently visualize each artifact type clearly. Visualized is a horizontal slice; the object photo shows the experimental setup specifically for the metal artifact scan. For all other artifact experiments, the metal object was removed.

**Figure 5 sensors-25-03076-f005:**
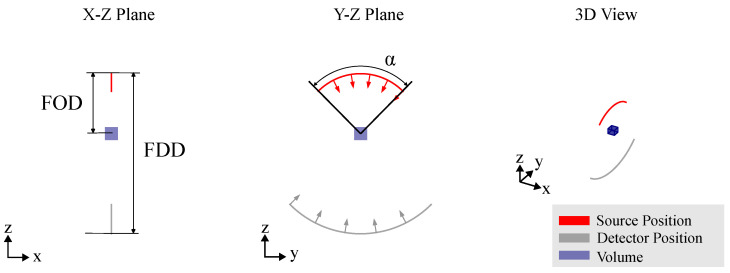
Circular CT trajectory. In contrast to a standard CT trajectory, the orientation and the center of the plane can be parameterized. This is limited due to kinematic and reachability constraints. Therefore, often only a limited angle α is reachable.

**Figure 6 sensors-25-03076-f006:**
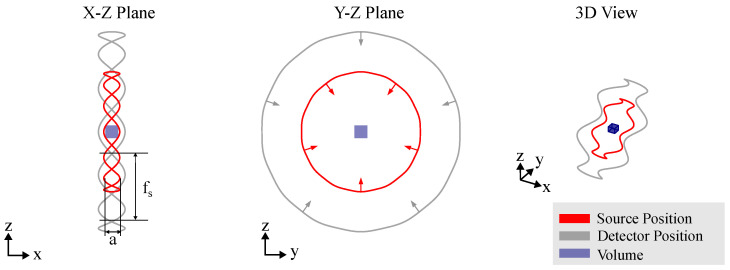
Circular CT trajectory with additional source movement. The flexibility of the system can be used to shift the source–detector trajectory in the normal direction of the CT trajectory plane. In this case, the height of the source is described by a sine wave with an amplitude *a* and the spatial period of the sine wave $f_{s}$, whereby FOD, FDD and the relative source–detector orientation remain constant.

**Figure 7 sensors-25-03076-f007:**
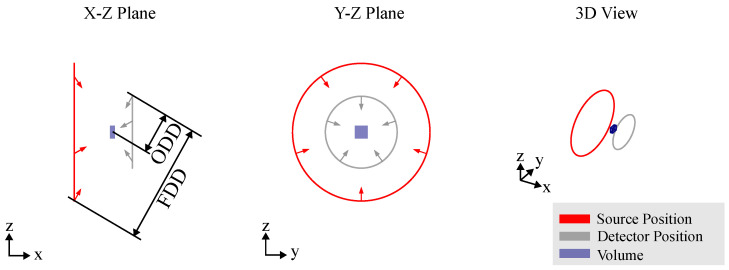
Circular Laminography trajectory. The X-ray source and detector move along circular paths on parallel planes, maintaining a constant distance from each other. During the scan, the X-ray beam remains perpendicular to the detector plane.

**Figure 8 sensors-25-03076-f008:**
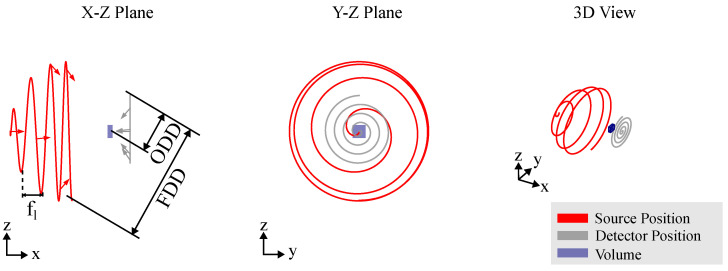
Spiral Laminography trajectory. The detector moves along a spiral path on a single plane, while the source position is determined by a line extending through the center of the measurement field and the current detector position. Throughout the scan, the FDD remains constant, while the FOD and ODD vary.

**Figure 9 sensors-25-03076-f009:**
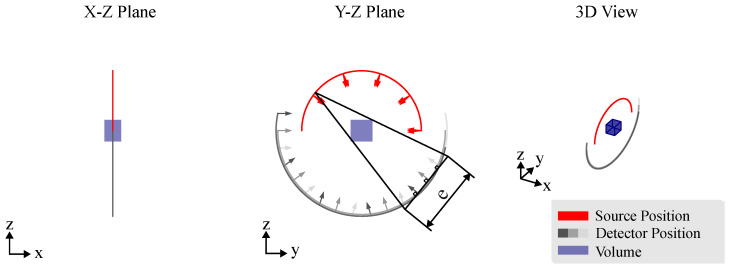
Projection stitching CT trajectory. For a single scan pose of any CT trajectory, the detector is moved within the imaging plane while keeping the source position fixed. This movement effectively increases the detector width to a size *e*, while maintaining the original scan geometry.

**Figure 10 sensors-25-03076-f010:**
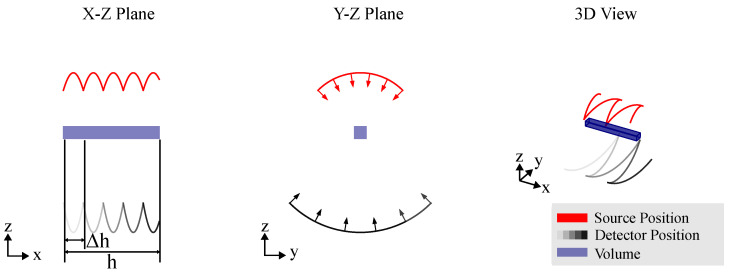
Reverse helical CT trajectory. The X-ray source moves on a helical path. This movement continues until the imaging angle α is reached, where the direction is reversed. As a result, the measurement field extends in the longitudinal direction by a factor of *h*.

**Figure 11 sensors-25-03076-f011:**
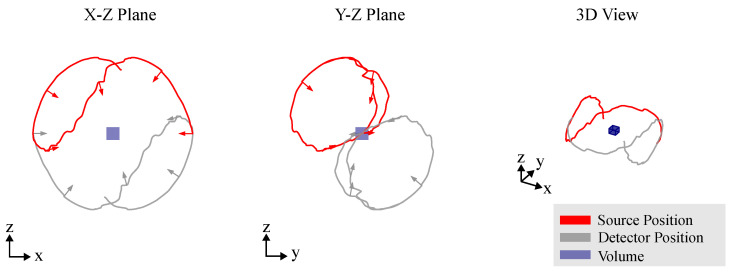
Arbitrary CT trajectory. These trajectories use all the flexibility of RoboCT systems. Source and detector position may not be on a parametrized path but an arbitrary set of scan positions. These trajectories are often the result of CT trajectory optimization algorithms.

**Figure 12 sensors-25-03076-f012:**
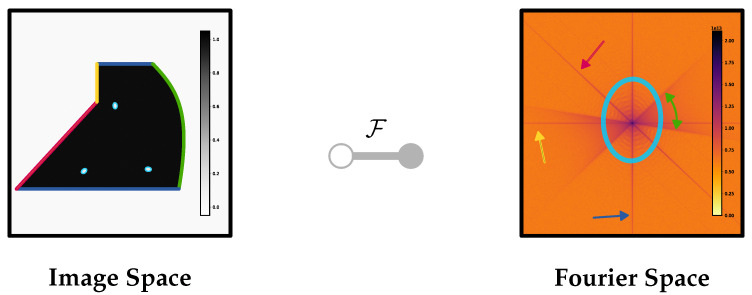
Transformation of an object from image space (**left**) into its frequency representation in Fourier space (**right**, the power spectrum is shown). The edges in the image space and their influence in Fourier space are highlighted. The figure concept is adapted from [22].

**Figure 13 sensors-25-03076-f013:**
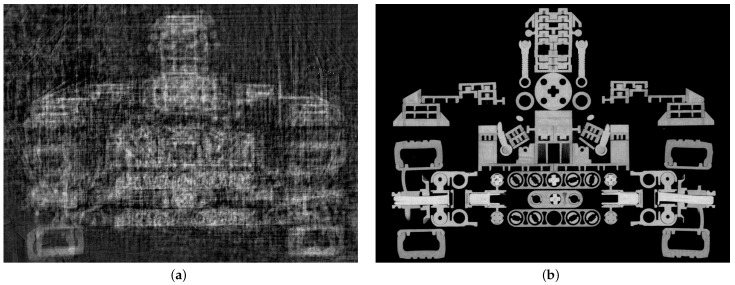
Reconstruction results of a LEGO car, scanned with the robot CT system of the Deggendorf Institute of Technology, shown in Figure 2, with incorrect geometric information (**left**) and with calibrated, correct geometric information (**right**). The reconstruction is based on a dataset of 1600 projections, each acquired by averaging three individual projections. In the slice shown, the specimen measures approximately 132 mm in width and 97 mm in height. The reconstruction was performed using the Feldkamp–Davis–Kress (FDK) algorithm with an isotropic voxel size of 0.278 mm. Furthermore, the observed reconstruction artifacts result from positioning errors of both robots—up to 2.7 mm—since a nominal model was assumed for demonstration purposes. (**a**) Reconstruction without any correction. (**b**) Reconstruction with a projection-wise geometry correction.

**Figure 14 sensors-25-03076-f014:**
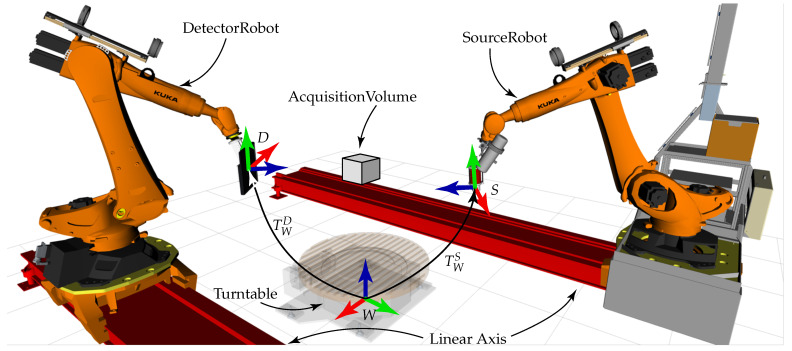
Digital twin of the twin robotic CT system at the Deggendorf Institute of Technology. The system is based on two KUKA KR 120 R2900 industrial robots, mounted on KUKA KL 4000 linear rails. The rotational stage used is of type WEISS CR 1000 C. The transformations $T_{W}^{D}$ and $T_{W}^{S}$ define the relation between the world coordinate frame *W* and *D*—Detector or *S*—Source, respectively. The precision and error of these transformations are the subject of calibration. An acquisition volume is depicted to showcase the approximate measurement volume of a single scan relative to the size of cells.

**Figure 17 sensors-25-03076-f017:**
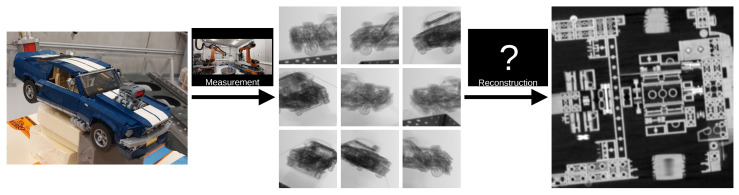
A LEGO 1967th Ford Mustang—a subset of measured projections and a reconstructed slice.

**Figure 18 sensors-25-03076-f018:**
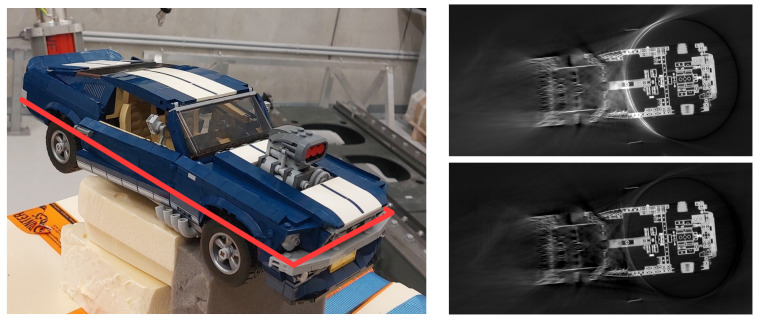
Comparison of an FBP (**top**) and SART (**bottom**) reconstruction of the complete object with projections covering not the complete object but only the car front with the car engine.

**Figure 19 sensors-25-03076-f019:**
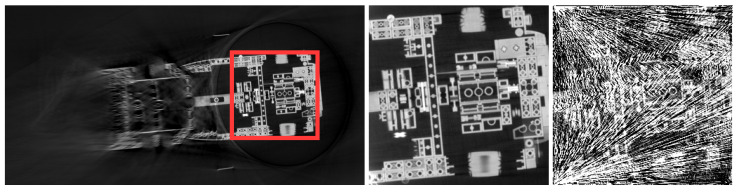
Outline of the region of interest (**left**) and the out-of-the-box reconstructions of the region of interest with FBP (**middle**) and SART (**right**). The FBP reconstruction is clear. The SART reconstruction contains artifacts.

## Data Availability

The study is a review article and does not involve new data.

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
