# Peer review of "RoboCT: The State and Current Challenges of Industrial Twin Robotic CT Systems"

_sensors, 2025, doi:10.3390/s25103076_

Round 1

Reviewer 1 Report

Comments and Suggestions for Authors

The article is an extensive review about robotic CT, and more precisely twin robotic CT.

The paper is well written and should be considered for publication. However, some revisions are necessary. The main focus of the paper is about twin robotic CT, which means dual robot CT systems. This should be indicated at the very beginning, as the term “twin” sometimes refer to “digital twins” which can be confusing. The difference between mono robot CT and dual robot CT is only briefly covered, while conventional CTsystems are too much described. My suggestion is to revise completely section 2 of the paper, and start with the definition of dual robotic CT, emphasizing the differences with mono robot CT. Conventional CT can be covered using one schema.

About the structure of the paper : the introduction explains the historical evolution of CT, and introduces very well the three challenges of robotic CT. Thus, it would be appropriate to have a section 2 devoted to definitions (which means defining what is robotic CT – replacing the actual section 2, + the definitions i.e. the actual section 3). Then, the three challenges as they are.

Indeed, the actual section 2 has a completely wrong title (the term actuator is not used in the whole text), and its content is not homogeneous : section 2.1 is too long, and contains too much self referencing of the authors, section 2.2 does not seem useful. Section 2.3 on the contrary should be developed with regards to the differences/complementarity with dual robot CT. This section can be fused with section 2.4 and just focusing on the strict necessity. Section 2.4 contains too many details which are in general repeated again in section 4. At present all the figures 1, 5-6-7-8 contain the same information : may be a table with associated references could replace the majority of them, and keep may be just one to give an idea of the dimensions. Then, figures 9 to 16 resemble a catalog which is not relevant even if you want to show the interest of robot CT. So, all the actual section 2 should be simplified and focused on the specificities of robotic CT without going in the details that are considered in the following sections. Just say that large objects can be scanned, that it is more versatile etc…in a concise way.

The rest of the paper is very clear, and all the suggestions for revision are included in the pdf file.

Other remarks concerns :

  • Some figures are missing, some are not informative (see the commented file),
  • references : along the text there is a change of style when citing the references (either by name or by number), and in the final list some references are incomplete.
  • the term “chapter” looks like the paper comes from a PhD study, which might be the case, but for a scientific review, may be the term “section” would be more appropriate (unless this is the right term in the provided text model).

Reviewer 2 Report

Comments and Suggestions for Authors

Please find the remarks in the attached word file.

Reviewer 3 Report

Comments and Suggestions for Authors

1. This review uses a lot of real pictures, which makes it look like a news report. It is recommended to replace it with a schematic diagram of the technical system to improve the academic nature of the review.
2. The author should give a clear mathematical paradigm for the classification of RoboCT, instead of a vague description.
3. there are too few formulas in the full text about the principle description of different RoboCTs, which leads to the lack of theoretical and academic nature of the REVIEW.
4. CT has made great progress in the field of medical diagnosis, and some cutting-edge CT technologies and applications have emerged, which the authors should add appropriate descriptions in the citation, e.g. Reduction of Compton Background Noise for X-ray Fluorescence Computed Tomography with Deep Learning, Computer-aided diagnosis and staging of pancreatic cancer based on CT images, Differentiating benign and malignant parotid gland tumors using CT images and machine learning algorithms.
5. Hot spots of RoboCT research and future research directions that can be carried out should be highlighted.
6. Different types of RoboCT should be given specific quantitative characteristic indexes, and there is no table in the whole paper, which is not quantitative enough.

Comments on the Quality of English Language

None

Round 2

Reviewer 3 Report

Comments and Suggestions for Authors

The authors have addressed my concerns well.

Comments on the Quality of English Language

None